# Attentional amplification of neural codes for number independent of other quantities along the dorsal visual stream

Elisa Castaldi[1]*, Manuela Piazza[2], Stanislas Dehaene[1], Alexandre Vignaud[3], Evelyn Eger[1]

[1]Cognitive Neuroimaging Unit, CEA DRF/JOLIOT, INSERM, Université Paris-Sud, Université Paris-Saclay, NeuroSpin Center, Gif-sur-Yvette, France; [2]Center for Mind/Brain Sciences, University of Trento, Trento, Italy; [3]UNIRS, CEA DRF/JOLIOT, Université Paris-Saclay, NeuroSpin Center, Gif-sur-Yvette, France

**Abstract** Humans and other animals base important decisions on estimates of number, and intraparietal cortex is thought to provide a crucial substrate of this ability. However, it remains debated whether an independent neuronal processing mechanism underlies this 'number sense', or whether number is instead judged indirectly on the basis of other quantitative features. We performed high-resolution 7 Tesla fMRI while adult human volunteers attended either to the numerosity or an orthogonal dimension (average item size) of visual dot arrays. Along the dorsal visual stream, numerosity explained a significant amount of variance in activation patterns, above and beyond non-numerical dimensions. Its representation was selectively amplified and progressively enhanced across the hierarchy when task relevant. Our results reveal a sensory extraction mechanism yielding information on numerosity separable from other dimensions already at early visual stages and suggest that later regions along the dorsal stream are most important for explicit manipulation of numerical quantity.
DOI: https://doi.org/10.7554/eLife.45160.001

*For correspondence:
elisa.castaldi@gmail.com

**Competing interests:** The authors declare that no competing interests exist.

## Introduction

One largely debated theme in cognitive neuroscience is how the human brain developed the ability to perform mathematics. While mathematical skills certainly rely on the interplay of a wide range of cognitive functions (*De Smedt et al., 2013*; *Fias, 2016*; *Iuculano and Menon, 2018*), an influential theory in the field proposes that a necessary prerequisite to develop such a sophisticated uniquely human ability resides in the 'number sense' (*Dehaene, 1997*). This is a phylogenetically ancient competence that enables humans and other animals to assess and mentally manipulate the approximate number of objects in sets. In humans the precision of the number sense (or 'numerical acuity', typically measured by visual number discrimination) sharpens with age and with the acquisition of formal mathematical education (*Piazza et al., 2013*), and correlates with arithmetical skills throughout the life-span (*Halberda et al., 2008*; *Libertus et al., 2011*; *Libertus et al., 2013*; *Chen and Li, 2014*; *Anobile et al., 2016a*; *Anobile et al., 2018*). Deviations from the typical developmental trend of numerical acuity can be a symptom of developmental dyscalculia (*Piazza et al., 2010*), a neurodevelopmental disorder that causes specific mathematical learning difficulties.

The neural substrate subtending this sense of numerical quantity is thought to be shared across species and has been linked to a network of areas in the frontal and parietal cortices sensitive to changes in numerosity since very early in life (*Izard et al., 2008*; *Hyde and Spelke, 2011*; see for reviews: *Cantlon, 2012*; *de Hevia et al., 2017*). In these areas electrophysiological recordings in monkeys identified single neurons tuned to specific numerosities of visual arrays (*Nieder et al.,*

**eLife digest** Numbers and the ability to count and calculate are an essential part of human culture. They are part of everyday life, featuring in calendars, computers or the weekly shop, but also in some of humanity's biggest achievements: without them the pyramids or space travel would not exist. A precursor of sophisticated mathematical skill could reside in a simpler mental ability: the capacity to assess numerical quantities at a glance. This 'number sense' appears in humans in early childhood and it is also present in other animals, but it is still poorly understood.

Brain imaging techniques have identified the parts of the brain that are active when perceiving numbers or making calculations. As techniques have advanced, it has become possible to resolve fine differences in brain activity that occur when people switch their attention between different visual tasks. But how exactly does the human brain process visual information to make sense of numbers? One theory suggests that humans use visual cues, such as the size of a group of objects or how densely packed objects are, to estimate numbers. On the other hand, it is also possible that humans can sense number directly, without reference to other properties of the group being observed.

Castaldi et al. presented twenty adult volunteers with groups of dots and asked them to focus either on the number of dots or on the size of the dots during a brain scan. This approach allowed the separation of brain signals specific to number from signals corresponding to other visual cues, such as size or density of the group. The experiment revealed that brain activity changed depending on the number of dots displayed. The signal related to number became stronger when people focused on the number of dots, while signals related to other properties of the group remained unchanged. Moreover, brain signals for number were observed at the very early stages of visual processing, in the parts of the brain that receive input from the eyes first.

These results suggest that the human visual system perceives number directly, and not by processing information about the size or density of a group of objects. This finding provides insights into how human brains encode numbers, which could be important to understand disorders where number sense can be impaired leading to difficulties learning math and operating with numbers.

DOI: https://doi.org/10.7554/eLife.45160.002

---

*2002*; *Nieder and Miller, 2004*; *Roitman et al., 2007*; *Nieder, 2016*) and fMRI studies in humans found activation in these areas to be modulated during quantity perception as well as during calculation (for reviews see: *Arsalidou and Taylor, 2011*; *Eger, 2016*; *Piazza and Eger, 2016*). While the first imaging studies in humans were limited by the low spatial resolution and univariate subtraction-based analyses, fMRI adaptation and multivariate pattern analysis methods provide higher sensitivity to finer-scale activity differences (*Kourtzi and Grill-Spector, 2005*; *Norman et al., 2006*; *Tong and Pratte, 2012*). These methods allowed researchers to study the representation of individual numbers by recording the distance-dependent signal release from adaptation (*Piazza et al., 2004*), or reading out patterns of number-related activity across multiple voxels of the frontal and parietal cortex (*Eger et al., 2009*). Moreover, population-receptive field mapping (pRF) methods identified individual locations tuned to specific numerosities arranged in spatially organized maps in the parietal cortex (*Harvey et al., 2013*).

While these earlier findings mostly pointed at the key role of parietal and frontal areas in numerical representation, some recent studies found that it is possible to decode the number of items seen by the subjects from the fMRI activity patterns in early visual areas (*Bulthé et al., 2014*; *Eger et al., 2015*; *Bulthé et al., 2015*; *DeWind et al., 2019*, but see *Castaldi et al., 2016*). Moreover, spatially organized numerosity maps were recently claimed to extend to the occipital cortex (*Harvey and Dumoulin, 2017a*) and early ERP components compatible with generators in early visual areas responded to variations in the numerosity of visual arrays (*Park et al., 2015*; *Fornaciai et al., 2017*; *Fornaciai and Park, 2017*).

Several properties characterizing numerosity perception, such as being ratio-dependent (Weber's law) and being susceptible to adaptation, led some authors to suggest that number is a 'primary' visual property of the image that is directly perceived through specialized and dedicated mechanisms (*Burr and Ross, 2008*; *Ross, 2010*; *Anobile et al., 2016b*). However, in spite of dedicated

efforts on modeling the extraction of numerosity from the visual image (*Dehaene and Changeux, 1993*; *Verguts and Fias, 2004*; *Dakin et al., 2011*; *Stoianov and Zorzi, 2012*; *Morgan et al., 2014*), the detailed neural processing mechanisms used by the brain to arrive at a representation of numerosity from the visual input remain little understood, and much less understood than the ones for other basic visual features such as orientation, colour, motion, etc. Numerosity is a notoriously difficult feature to study since changes in numerosity tend to be associated with changes in other quantitative features of the sets during natural viewing conditions (e.g., more items tend to occupy a larger area, or be spaced more densely), and it appears impossible to control for all of these associated quantities at the same time. For this reason, in spite of a large body of behavioral and neuroscientific work on this topic, it still remains debated whether the available evidence supports a sensory extraction mechanism directly sensitive to numerosity. Some have argued instead that numerosity might be judged indirectly by weighing a combination of other, non-numerical, quantitative features of the stimuli (*Gebuis and Reynvoet, 2012*; *Gebuis et al., 2014*; *Leibovich et al., 2016a*). For example, numerosity can be mathematically defined as the product of density (number of items per unit of area) by field area; or by the total surface area divided by mean item size. Thus, decisions on numerical quantity could be taken merely indirectly, on the basis of representations of these non-numerical properties, without numerosity being encoded directly by perceptual systems.

While this possibility is interesting, several behavioral findings argue against it: (1) the discrimination of numerosity and of one often correlated non-numerical feature (item density) follow different psychophysical laws (*Anobile et al., 2016b*), and (2) at least for relatively small numbers of not too densely spaced items, perceptual thresholds for numerosity discrimination are typically much smaller than the ones predicted from the thresholds for density and field area together (*Cicchini et al., 2016*), making it unlikely that estimates of numerosity are based on the latter. For what concerns the neuronal level, a few recent studies have started to directly quantify the effects of non-numerical dimensions of non-symbolic numerical stimuli (e.g. *Park et al., 2015*; *Fornaciai et al., 2017*; *Harvey and Dumoulin, 2017b*; *Fornaciai and Park, 2018*; *DeWind et al., 2019*). Those studies found that activity in earlier (occipital) or later (parietal) brain regions appeared to be linked to the numerical content of sets after taking into account effects of certain non-numerical dimensions. However, they mostly only considered the effect of one non-numerical variable at the time and compare it to that of number, without taking into account effects explained by all relevant non-numerical dimensions together. Thus, it still remains unclear to what extent activity evoked by non-symbolic numerical stimuli within early and later regions can be explained by a mechanism that encodes numerosity in itself, or by the ensemble of responses to the different non-numerical dimensions of the stimuli.

Here, we implement a new approach to separate brain signals related to numerical and non-numerical quantities and test for a neuronal mechanism directly sensitive to the numerosity of visual sets along the dorsal visual stream hierarchy. We propose that the following signatures would advocate for the existence of such a mechanism:

First, information on numerosity should be detectable in regional activity patterns after multiple important non-numerical quantities are simultaneously (and not only individually) taken into account. Second, and importantly, it should be possible to selectively amplify this numerical information depending on whether the numerical dimension of the stimuli is task relevant, similar to the attentional amplification that has been previously shown for other task-relevant primary features, such as orientation, contrast, color, direction etc. (*Jehee et al., 2011*; *Ester et al., 2016*). If a brain area encodes numerical information in a way that is separable from associated non-numerical dimensions, tasks involving selective attention to number should enhance the information about numerosity, without affecting the level of information on associated non-numerical dimensions. Thus, we propose that the presence of such independent attentional amplification is a key criterion in order to identify which brain areas explicitly encode information on numerosity.

On the contrary, if activity patterns could be entirely accounted for by the combination of responses to multiple non-numerical dimensions of the stimuli, no information specifically related to number should be found in the patterns of activity once accounting for the other (non-numerical) dimensions simultaneously. Furthermore, if numerosity was not directly encoded but only indirectly inferred from percepts of non-numerical properties, attentional enhancement should not occur for signals related to numerosity, but if anything, only for other properties (e.g., density and field area) that can jointly define it.

To test these predictions, we created a novel stimulus space to disentangle the contribution of numerical and non-numerical dimensions to brain activity patterns, and designed a task where attention is selectively directed towards either of two orthogonal quantitative dimensions of the visual array (number or item size). We exploited the enhanced sensitivity achieved by fMRI at ultra-high field (7 Tesla) and specific multivariate pattern analyses to simultaneously model and separate the contributions of numerical and different non-numerical quantities to fine-scale activity patterns within multiple regions defined by a probabilistic atlas based on visual topography.

## Results

We scanned twenty healthy adult volunteers while they performed two tasks on arrays of dots varying orthogonally in numerosity (6, 10, or 17 items), average item size (0.04, 0.07, or 0.12 visual square degrees - vd$^2$) and total field area (44 or 20 vd$^2$) (*Figure 1A*). Participants alternated between a 'number' and a 'size' task in different blocks: during the 'number' blocks they had to direct attention to the numerosity of each sample stimulus and keep it in memory for comparison with an occasionally following match stimulus, while during 'size' blocks they performed the equivalent task on the average item size of the arrays (*Figure 1B*). When a match stimulus appeared (indicated by a change in color of the fixation point), participants had to decide whether the match stimulus was larger or smaller on the attended dimension than the previous sample held in memory and to respond by button press.

### Behavioral performance and univariate fMRI activation effects

Response accuracies for comparison of match stimuli were overall high and not significantly different across tasks (86% for the number task and 85% for the average size task, t(19) = 0.46, p = 0.65), suggesting that subjects attended to the correct stimulus dimension and the difficulty was on average successfully matched across tasks (*Figure 2A*).

We started the analysis of the functional imaging data by evaluating overall regional activation effects during both tasks. Surface-based random-effects group analysis identified similar bilateral activations in the occipito-parietal and frontal cortex during both tasks for sample stimuli against the implicit baseline where participants were just looking at the fixation point with blank screen and without performing any task (*Figure 2B and C*, thresholded at p < 0.001 uncorrected). In both tasks the activity covered a wide occipito-parietal area starting from the superior occipital and transverse occipital sulci and extending throughout the intraparietal sulcus (IPS) up to the post-central sulcus.

The frontal activity mainly covered the superior frontal gyrus. The direct contrast of sample stimulus-related activity during the number versus the size task revealed no area with significantly stronger activation for either of the two, despite the uncorrected significance threshold (*Figure 2D*). Altogether, these results suggest that task difficulty was successfully matched and that under these conditions attending to different quantitative dimensions leads to equivalent overall activation of the brain regions involved in the task. Differences in overall activation level can therefore not confound the following more specific results on the within-dimension discriminability of quantitative features.

### Multivariate fMRI Pattern Analyses

#### Read-out of sample numerosity is modulated by task

Given that the whole brain univariate contrasts had confirmed equivalent activations across the two tasks, we further investigated, using multivariate classification, what was the degree of discriminability of activity patterns evoked by different sample numerosities across different regions of the dorsal visual stream and during the number and size task. In each subject we identified several regions of interest (ROIs) derived from a surface-based probabilistic atlas based on visual topography (*Wang et al., 2015*), (*Figure 3A*). Within each region, we used an equivalent number of most activated voxels (in the orthogonal contrast 'all sample stimuli > baseline') to train and test multivariate classifiers to discriminate between numerosities for each task (for the ability of the classifier to discriminate task see *Figure 3—figure supplement 1* and *Supplementary file 3* of the Supplementary Material). *Figure 3B* shows the across-subject overlap map for the included voxels which mainly highlight the foveal portion of the different ROIs, in line with the central presentation of the dot arrays. We first compared decoding accuracies in three large regions corresponding to early,

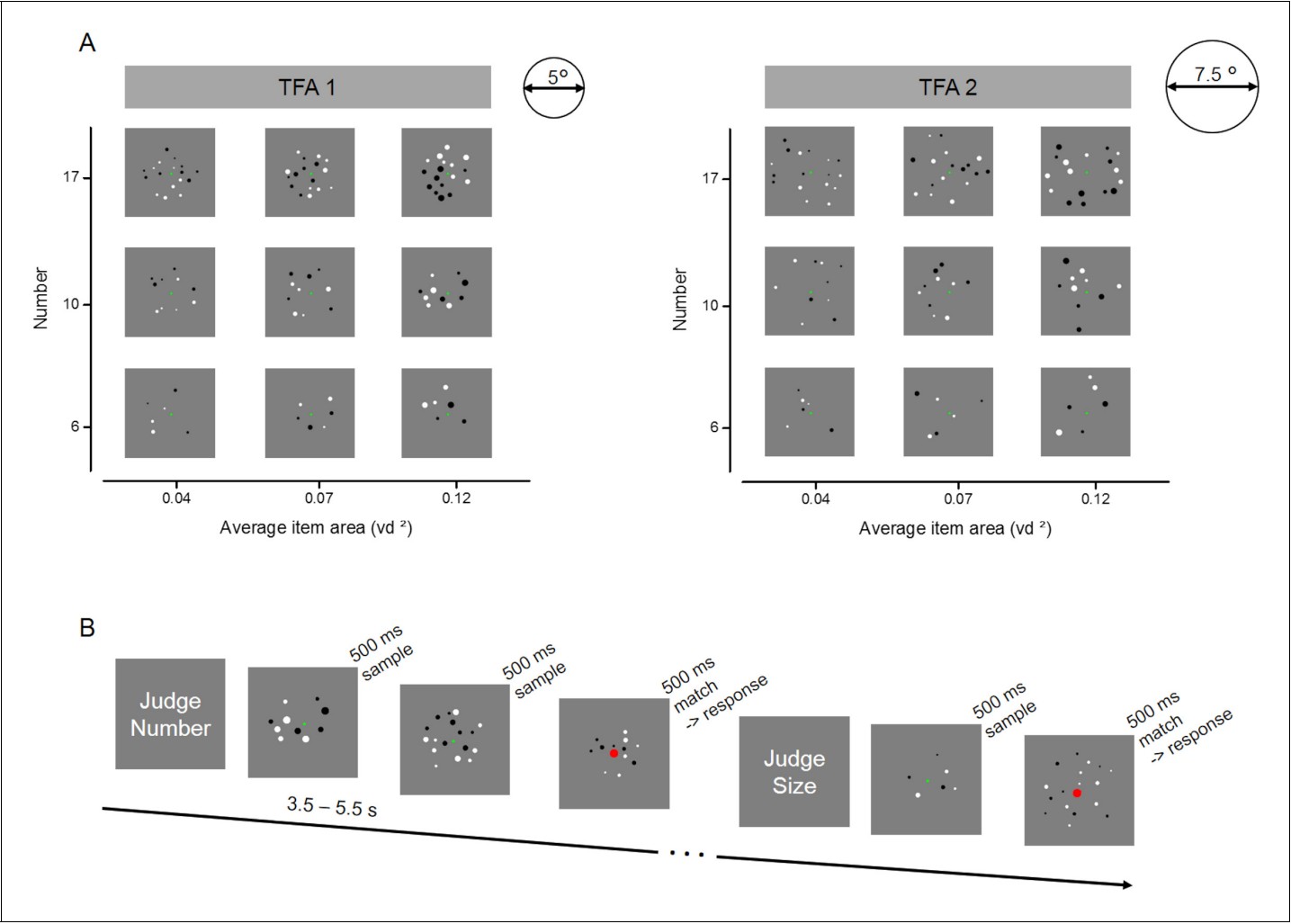

**Figure 1.** Stimulus set and design for the fMRI experiment. (**A**) Example of the full set of stimulus conditions. Arrays of six, ten or seventeen dots were created with three average item areas (0.04, 0.07 and 0.12 visual degree[2]) and displayed within two total field areas, enclosed by imaginary circles of 5˚ (TFA 1) and 7.5˚ (TFA 2) diameter. (**B**) Illustration of the trials' temporal presentation and paradigm during scanning. At the beginning of each block, written instructions informed participants about the dimension to attend: either the numerosity or the average size of the dots arrays. Participants were instructed to keep in memory the relevant dimension of each sample trial until the following trial was shown (after a variable time interval of 3.5–5.5 s). The color of the fixation point in the upcoming trial provided further instruction: if it remained green, participants had to update their memory with the new stimulus (new sample trial), while if it turned red, participants had to compare the current stimulus (match trial) with the one kept in memory, and to indicate by button press whether the match stimulus was larger or smaller than the sample on the attended dimension. After the response a new sample stimulus appeared after at least 8 s. FMRI analyses focused on activity evoked by sample stimuli only.
DOI: https://doi.org/10.7554/eLife.45160.003

The following figure supplement is available for figure 1:

**Figure supplement 1.** Additional illustration of stimulus set.
DOI: https://doi.org/10.7554/eLife.45160.004

intermediate and higher-level areas (including areas from V1 to V3, from V3AB to V7 and from IPS1 to IPS5, respectively). Then, to track the presence of information discriminative of numerosity across the dorsal visual stream more in detail, we further compared the classification accuracies across seven contiguous ROIs from V1 up to IPS345 (for results concerning the intraparietal sulcus excluding those regions defined by visual topography see *Figure 3—figure supplement 2A* and *Supplementary file 4a and 4b* in Supplementary Material). *Figure 3C* shows the performance of the classifiers trained to discriminate between different numerosities as a function of task. Overall, the presented sample numerosity could be decoded significantly above chance in all the ROIs and

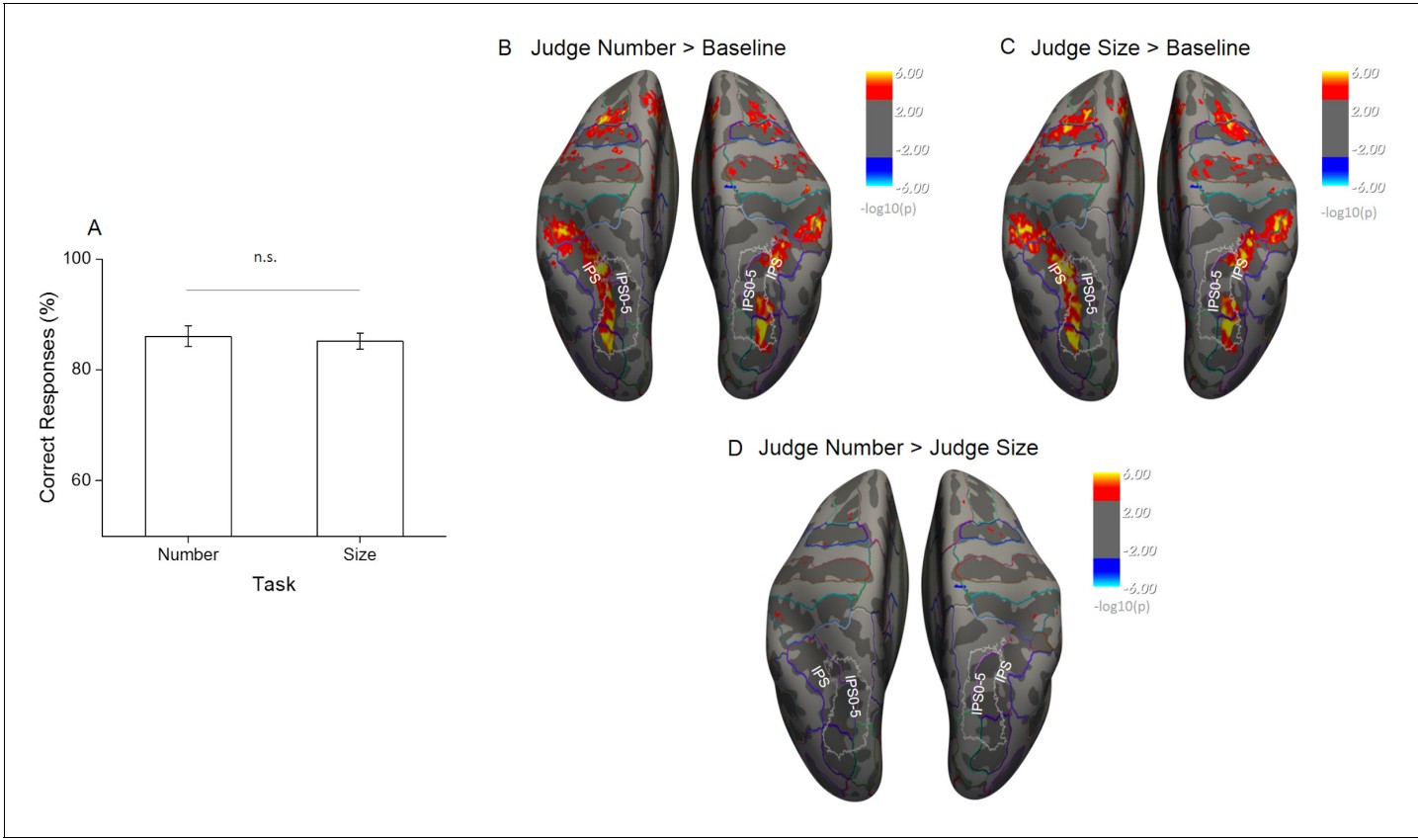

**Figure 2.** Behavioral performance during scanning and univariate effects of task. (**A**) The percentage of correct responses to match stimuli for the two tasks performed during scanning shows that task difficulty was successfully matched (n = 20, mean ± standard error of the mean (SEM)). (**B–D**) Statistical results obtained from the surface based group analysis (n = 20). The maps show the activation elicited for all sample trials during the number task (**B**) and the size task (**C**) when contrasted against the implicit baseline and against each other (**D**). Activation maps are thresholded at p < 0.001, uncorrected for multiple comparison, and displayed on Freesurfer's fsaverage surface with colored outlines identifying the major anatomical sulci and gyri based on the Destrieux Atlas (*Fischl, 2004*) and white outlines the field maps IPS0-5 based on visual topography (*Wang et al., 2015*).

DOI: https://doi.org/10.7554/eLife.45160.005

during both the number and size task, however with important differences. When explicitly attending to numerosity, the classification accuracy gradually increased across the dorsal stream (starting to be enhanced from intermediate areas, specifically from V3AB on), and was highest in parietal areas. During the size task, when attention was not explicitly directed towards the numerical aspect of the stimuli, the different numerosities were still decodable, however the classification accuracies were reduced in intermediate and higher regions, while they remained almost unchanged in early visual areas (specifically in V1, V2 and V3).

The task-driven modulation of decoding accuracies across the three major ROIs is confirmed by a significant interaction between ROI and task (F(1.69,32.11) = 9.81, p = 0.0008). For the number task, the classification accuracy progressively increased from the early visual areas (slightly above 60%) to intermediate and higher level regions where it reached almost 70% correct. Post-hoc tests showed that the classification accuracy increase in intermediate and higher areas with respect to early areas was very close to or clearly significant (p = 0.075, Cohen's d = 0.52, and p = 0.028, Cohen's d = 0.57 respectively). During the size task, the classification accuracy in the intermediate and higher regions dropped down to 61% and 60% respectively (yet remaining highly significantly above chance in both cases, see p-values in *Supplementary file 1*). The change in classification accuracy across tasks was highly significant both for the intermediate and higher areas, (p = 0.001, Cohen's d = 0.88 and p = 0.00001, Cohen's d = 0.98). On the other hand, the classification accuracy in the early visual areas remained nearly constant (62%) and was not significantly modulated by task (p = 0.5).

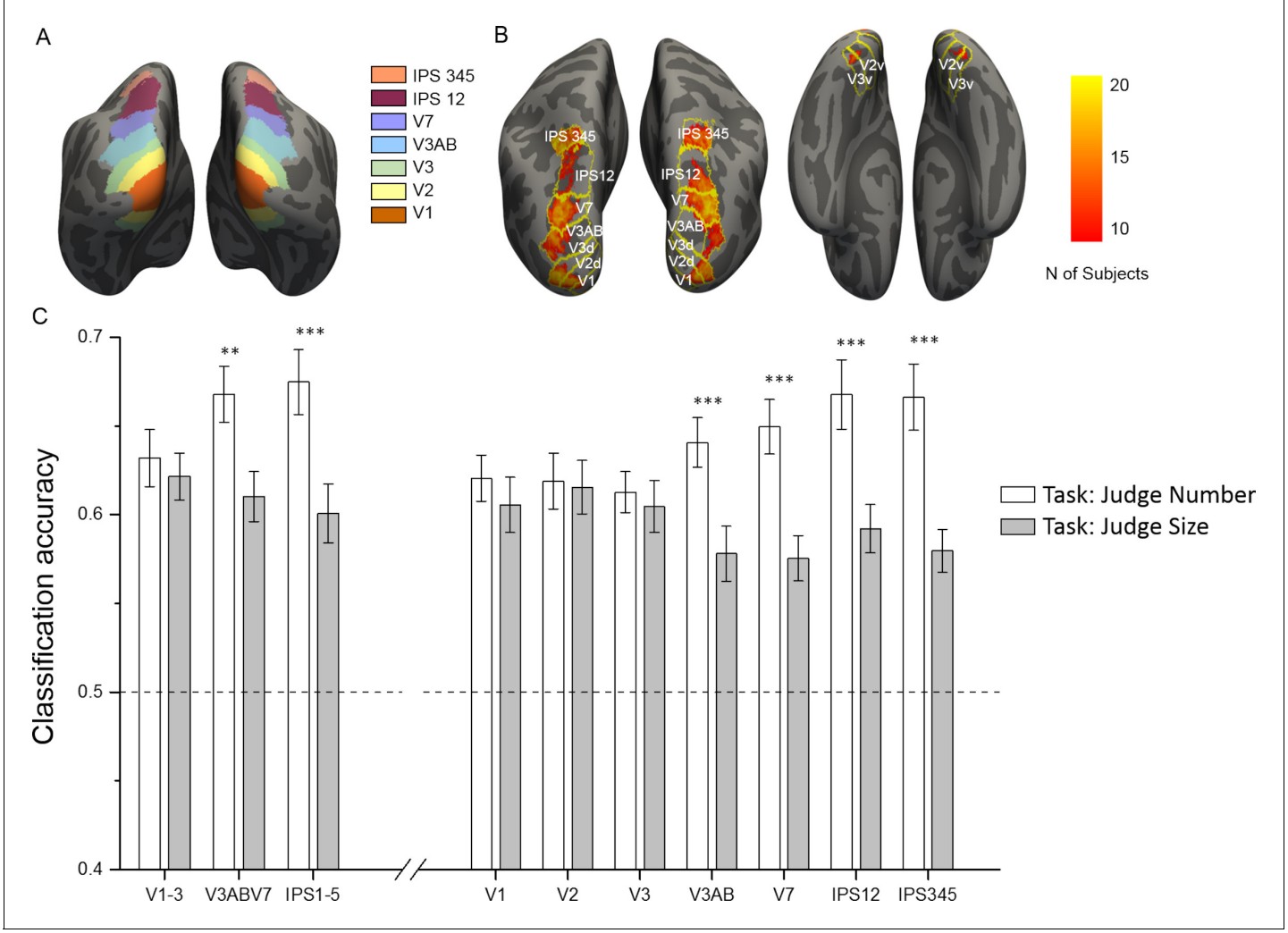

**Figure 3.** ROI localization and results of multivariate classification for discrimination between numerosities as a function of the task. (A) Color-coded ROIs defined by the probabilistic atlas are shown on the inflated brain template. (B) Across-subject overlap map of the most activated voxels in the contrast all sample > baseline. For each subject the most activated voxels were selected from each ROI (outlines) and hemisphere and the color map shows the number of subjects for which a given location was selected. (C) Sample numerosities could be classified significantly above chance across all the combined (left side) and individual (right side) ROIs, both during the number (white bars) and size (gray bars) task. The classification performance is strongly modulated by task only in the intermediate and higher-level ROIs, starting from V3AB on, but not in the early areas (V1, V2 and V3). Results show mean classification accuracy across subjects (n = 20) ± standard error of the mean (SEM). Stars mark the difference across tasks, not against chance level (which is significant for all regions and tasks; see ***Supplementary file 1*** for statistical results).

DOI: https://doi.org/10.7554/eLife.45160.006

The following figure supplements are available for figure 3:

**Figure supplement 1.** Results of multivariate classification for discrimination between tasks.
DOI: https://doi.org/10.7554/eLife.45160.007
**Figure supplement 2.** Results for the ROI defined along IPS excluding IPS0-5.
DOI: https://doi.org/10.7554/eLife.45160.008

The significant interaction between ROI and task was confirmed when testing the seven individual regions (F(4.10,77.97) = 7.17, p = 0.00005). Although the post-hoc tests did not show significant differences in classification accuracies across individual ROIs, for the number task the decoding accuracies progressively increased across the visual hierarchy and varied from slightly above 60% in the primary visual areas (V1 = 62%, V2 = 62%, V3 = 61%) up to almost 70% in the intermediate and higher ROIs (V3AB = 64%, V7 = 65%; IPS12 = 67%, IPS345 = 67%). During the size task, decoding accuracies were much reduced in intermediate and higher regions (V3AB = 58%, V7 = 58%;

IPS12 = 59%, IPS345 = 58%, yet still significantly above chance in all ROIs, see p-values in *Supplementary file 1*), while they remained almost unchanged in the primary visual areas (V1 = 61%, V2 = 62%, V3 = 60%). Accordingly, the post-hoc tests indicated that the classification accuracy in individual regions significantly changed across tasks only from V3AB on (V1: p = 0.28, Cohen's d = 0.24; V2: p = 0.83, Cohen's d = 0.05; V3: p = 0.55, Cohen's d = 0.14; V3AB p = 0.0002, Cohen's d = 0.98; V7: p = 0.00003, Cohen's d = 1.20; IPS12: p = 0.00001, Cohen's d = 1.03; IPS345: p = 0.00005, Cohen's d = 1.26).

In sum, multivariate classification analyses revealed that the sample numerosity presented could be read out from brain activity patterns in all ROIs tested during both tasks, although accuracy was enhanced in mid-to-higher level but not in earlier regions when number was the attended feature. However, since in this analysis activations for all sample stimuli for a given numerosity were pooled together, the decoding performance obtained could still be partly driven by features other than numerosity per se.

## Multiple regression RSA to disentangle the contributions of quantitative dimensions

As a critical test of whether the representations of numerical and non-numerical features of the stimuli could be dissociated across the dorsal visual stream, we performed Representational Similarity Analysis (*Kriegeskorte, 2008*; *Kriegeskorte and Kievit, 2013*) which, unlike classification-based decoding, allows to assess the effect of multiple quantitative dimensions on activity patterns simultaneously. For each ROI and task, we obtained a neural representational dissimilarity matrix (neural RDM, *Figure 4A*) by computing the correlation distance between activation patterns for each possible pair of conditions. We then applied multiple regression analysis to test in how far the fMRI pattern dissimilarity structure could be explained by multiple predictor matrices reflecting the stimuli's dissimilarity along several important quantitative dimensions: numerosity, average item size, total field area, total surface area and density (*Figure 4B*). Of note, our design orthogonally manipulating numerosity, average item size and total field area ensured that numerosity was also partly decorrelated from density and total surface area (as shown by the correlation values in the Predictor Correlation matrix, *Figure 4B*), yet not completely (correlation between number and density predictors = 0.43; between number and total surface area predictors = 0.33). Correlations between predictors in a multiple regression lead to a reduction of the unique variance attributable to each one of them, and to a greater variability of the estimated betas. An estimation of variance inflation factors (VIF) for each predictor in our case revealed that these remained reasonably low (corresponding to 1.4874, 1.1957, 1.2048, 1.3238 and 1.4591 for number, average item size, total field area, total surface area and density, respectively). By using a multiple regression approach we capitalize on the fact that the resulting beta weights reflect only the part of the variance that each one of these stimulus descriptors uniquely explained in the pattern of activity of a given ROI on top of all the others. Thus, by entering numerical and non-numerical dimensions together into a multiple regression, a significantly above zero beta for number would imply that the numerical information is contributing to the pattern of activity within a given ROI, over and above the contribution of the other non-numerical quantitative dimensions.

*Figure 5* displays the results of the estimated beta weights for various ROIs separately for the number (*Figure 5A*) and size tasks (*Figure 5B*). Beta weights for the effect of number independent of the other dimensions (black triangles) were generally positive and progressively explained the activity patterns better when proceeding from lower to higher-level regions when task relevant. The evolution of the numerical information across the visual stream was attenuated during the size task, yet betas remained significantly above zero in all regions (see p-values in *Supplementary file 2*). Beta weights for the non-numerical dimensions (other shapes in *Figure 5*) were pronounced predominantly in the earlier visual areas and, importantly, they appeared to be not clearly affected by task.

## Quantitative dimensions are modulated by task across ROIs to different extent

To statistically test for differential modulation of the contribution of the different quantitative dimensions to activation patterns, beta weights were analyzed with repeated measure ANOVAs with ROI,

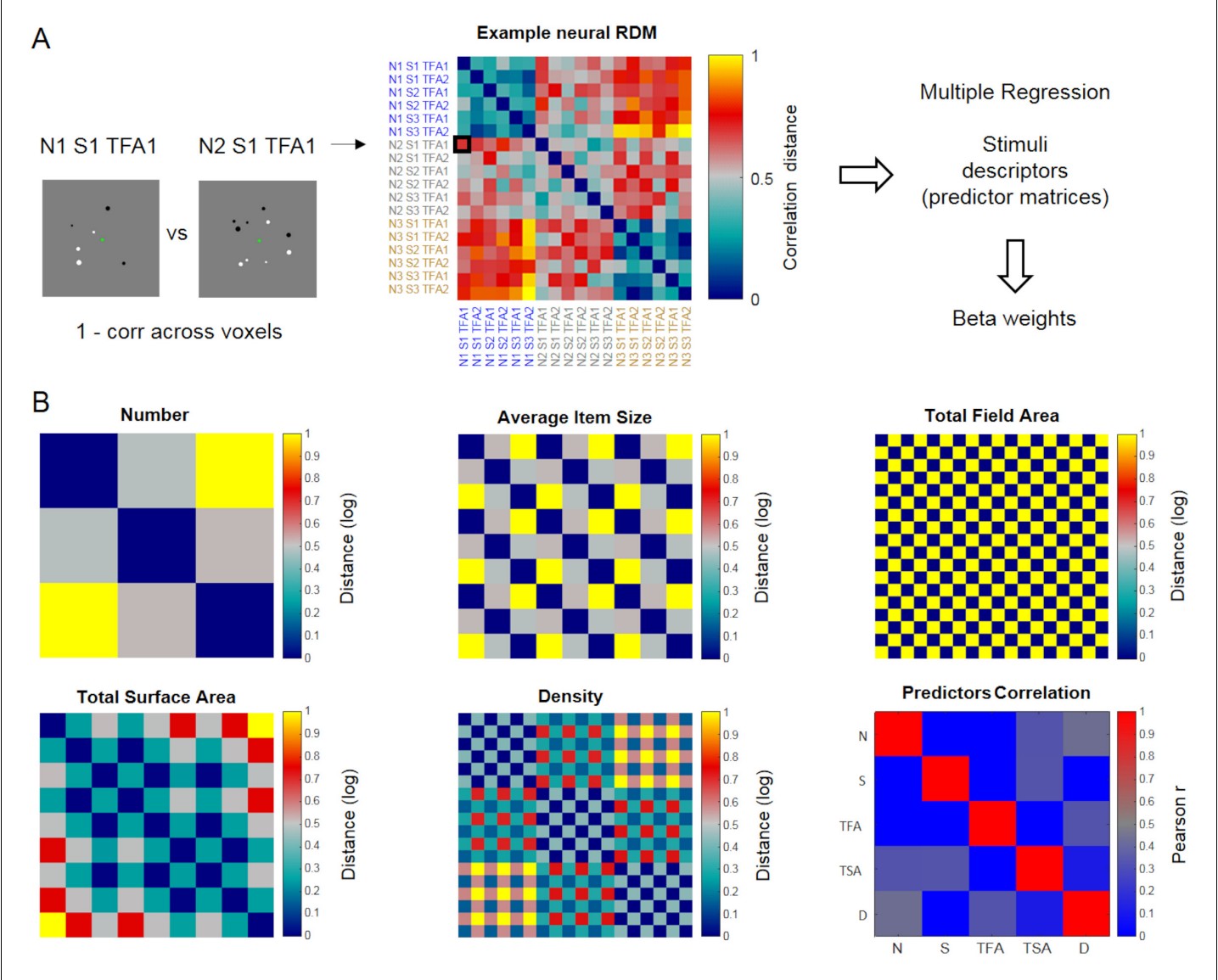

**Figure 4.** Schematic illustration of representational similarity analysis. Neural representational dissimilarity matrices (RDM) derived from fMRI were entered into a multiple regression where predictors corresponded to five matrices describing the dissimilarities across stimulus conditions along numerical and non-numerical dimensions. (**A**) Example neural RDM, quantifying the correlation distance (1 – Pearson correlation) between the patterns of activity elicited by all possible pairs of stimulus conditions across voxels within a given ROI (matrix scaled between 0 and 1 for visualization purposes). Each cell represents the correlation distance between activity patterns associated with a given pair of stimulus conditions (relatively lower values indicate more similar, and higher values more dissimilar patterns, respectively). The total of 18 conditions correspond to the combinations of 3 numerosities (N1, N2 and N3 corresponding to 6, 10 and 17 dots), three average item sizes (S1, S2 and S3 corresponding to small, medium and large average item sizes) and two total field areas (TFA1 and TFA2 corresponding to small and large total field area). The labels' colors follow changes along the numerical dimension. (**B**) The five dissimilarity matrices used as predictors in the multiple regression analysis represent the logarithmic distance between pairs of stimuli in terms of number, average item size, total field area, total surface area and density (all matrices scaled between 0 and 1 for visualization purposes). The correlation across these five predicted matrices is shown in the 'predictor correlation' matrix.

DOI: https://doi.org/10.7554/eLife.45160.009

task and dimension as factors. As for the classification analysis, we first focused on the three large regions corresponding to early, intermediate and higher-level areas and then further on individual ROIs from V1 up to IPS345 (for results concerning the intraparietal sulcus excluding those regions defined by the atlas based on visual topography see *Figure 3—figure supplement 2B* and *Supplementary file 4c*).

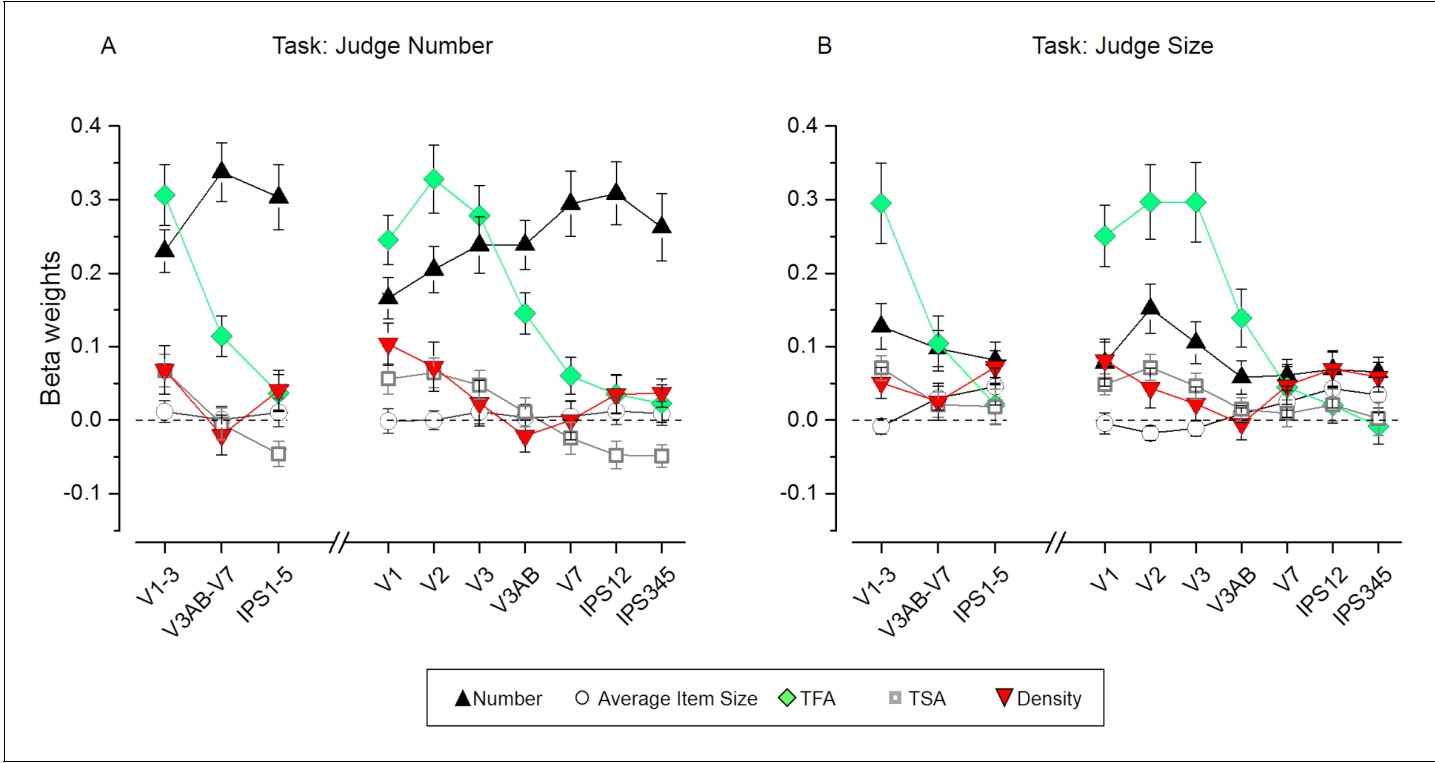

**Figure 5.** Results of the representational similarity analysis. Beta weights obtained from the RSA multiple regression analysis for number (black triangles), average item size (circles), total field area (TFA, diamonds), total surface area (TSA, squares) and density (red triangles) for the number (**A**) and size (**B**) task. While the fMRI pattern dissimilarity in early visual areas reflected contributions of multiple properties (TFA, density, TSA, but also number on top of these), when attending to number (**A**) the dissimilarity matrix for number increasingly better explained the fMRI pattern dissimilarity when progressing towards higher areas of the dorsal visual stream, where the contribution of non-numerical dimensions was smaller. The dissimilarity matrix for number however, contributed much less to explain neural dissimilarity in mid- and higher-level ROIs during the size task (**B**). The contribution of the non-numerical dissimilarity matrices remained mostly unaffected in most of the ROIs, with only a slightly enhanced contribution of the dissimilarity matrix for density which significantly contributed to explain the neural RDMs in higher areas during the size judgments. Data points show mean beta weights across subjects (n = 20) ± standard error of the mean (SEM). P-values testing the significance of the beta coefficients for each dimension and ROI are reported in **Supplementary file 1**.

DOI: https://doi.org/10.7554/eLife.45160.010

The following figure supplements are available for figure 5:

**Figure supplement 1.** Results of the representational similarity analysis for a model including only number, average item size and total field area as predictors.

DOI: https://doi.org/10.7554/eLife.45160.011

**Figure supplement 2.** Results of the representational similarity analysis for a model including all the non-numerical dimensions only (i.e. average item size, total field area, total surface area and density).

DOI: https://doi.org/10.7554/eLife.45160.012

The significant triple interaction between ROI, task and dimension confirmed that the beta weights estimated for the different dimensions were differently affected by task across ROIs (for the three large regions: $F_{(4.23, 80.40)} = 3.32$, $p = 0.01$; for the individual regions: $F_{(6.18, 117.38)} = 3.06$, $p = 0.007$). To identify which dimension was maximally driving this effect, we quantified the changes in beta weights across ROIs and tasks for each dimension separately.

## Effects of the numerical dimension

Beta values for number were the only ones showing a significant interaction between ROI and task, when comparing the three large subdivisions across the visual stream ($F_{(1.35, 25.62)} = 5.97$, $p = 0.015$). During the number task, betas for number were higher in intermediate and higher-level areas with respect to early visual areas (although only the former comparison was significant, $p = 0.04$, Cohen's d = 0.70). During the size task the betas for number were significantly lower

(significant difference across tasks in early: p = 0.007, Cohen's d = 0.78; intermediate: p = 0.000001, Cohen's d = 1.96; higher areas: p = 0.00001, Cohen's d = 1.43) and not different across regions.

When focusing on the seven individual ROIs, the interaction between ROI and task was significant $(F_{(2.04, 38.83)} = 5.29$, p = 0.009). Although post-hoc tests did not identify significant differences across ROIs, linear regression showed that the increase in beta weights for number across the dorsal visual stream was significant during the number task only $(F_{(1,5)} = 14.23$, p = 0.01, $R^2 = 0.74$), while during the size task betas for number were much more homogenous across ROIs $(F_{(1,5)}=2.37$, p = 0.18, $R^2 = 0.32$). Indeed the difference in beta weights between the number and size task was only minor or not significant in V1 and V2, more pronounced in V3, and highly significant from V3AB on (difference across tasks: V1: p = 0.025, Cohen's d = 0.64; V2: p = 0.13, Cohen's d = 0.37; V3: p = 0.001, Cohen's d = 0.91; V3AB p = 0.000001, Cohen's d = 1.44; V7: p = 0.000008, Cohen's d = 1.54; IPS12: p = 0.000001, Cohen's d = 1.56; IPS345: p = 0.000112, Cohen's d = 1.28).

## Effects of the non-numerical dimensions

Different from number, beta weights estimated for the non-numerical dimensions were not modulated by task (no significant interaction between ROIs and task, no significant main effect of task) for any of the dimensions.

Independent of the task, total field area best explained activity patterns in early visual areas, while its contribution was reduced when proceeding through intermediate to higher-level areas (significant main effect of ROIs: $F_{(1.23, 23.29)}=35.24$, p = 0.000002; significant differences in beta weights between primary and intermediate or higher-level ROIs: p = 0.000155, Cohen's d = 1.24, p = 0.000008, Cohen's d = 1.80, respectively). Beta values were highly significantly modulated also across the different individual ROIs (main effect of ROIs: $F_{(2.11, 40.12)} = 32.27$, p $< 10^{-5}$). Indeed, activity patterns in V1, V2 and V3 were explained equally well by total field area and better than intermediate and higher regions, starting from V3AB on (all p < 0.01 at least).

Total surface area also most strongly modulated pattern dissimilarity in early visual areas. The significant main effect of ROI $(F_{(1.45, 27.63)} = 16.61$, p = 0.000078) and the following post-hoc tests showed that beta values for this dimension in the early visual areas were significantly higher than those estimated for the intermediate (p = 0.000475, Cohen's d = 0.76) and higher-level (p = 0.000943, Cohen's d = 1.30) ROIs, independent of the task. Beta weights for total surface area were comparable in V1, V2 and V3 (no significant difference across these ROIs) and significantly higher than those of the others ROIs starting from V3AB/V7 on (significant main effect of ROI: $F_{(3.13, 59.42)} = 13.27$, p = 0.000001, comparisons across regions: all p < 0.01 at least).

Density modulated early visual areas during the number task and both earlier and higher-level areas during the size task. The main effect of ROI was significant $(F_{(1.41, 26.72)} = 4.05$, p = 0.04), but additional post-hoc tests did not reveal any significant difference across the three large ROIs. Also at the level of individual regions the main effect of ROI was significant $(F_{(2.55, 48.55)} = 4.15$, p = 0.01) and the strongest difference across ROIs emerged when comparing the lowest beta weights estimated in V3AB with those obtained in V1 (p = 0.003, Cohen's d = 1.13) and V7, IPS12 and IPS345 (p = 0.03, Cohen's d = 0.20, p = 0.01, Cohen's d = 0.53, p = 0.002, Cohen's d = 0.64).

Surprisingly, effects due to average item size could not be detected in any of the ROIs tested.

In sum, while early visual areas contained independent information on multiple quantitative properties of which some explained more variance than numerosity, all regions were modulated to some extent by numerical distance over and above what was explainable by the non-numerical dimensions. Moreover, importantly, explicitly directing attention to number did enhance the representation of numerical information and did so selectively, without altering the representations of non-numerical quantities. Finally, although present starting from the earliest stages of visual analysis, the numerical information at this level was only to a minor extent modulated by task and the greatest contribution to explicit manipulation of numerical quantity was found in intermediate and higher-level regions.

## Models with reduced number of predictors

Overall, the model described in the previous sections, which takes into account multiple numerical and non-numerical dimensions simultaneously, seems to us the most appropriate way to address our main question, which concerns the contribution of numerosity to activity patterns above and beyond

what can be explained by the non-numerical dimensions. However, since some correlations were present between the predictors for the different quantitative dimensions in our original model, we further explored in how far results might differ when modeling either only the orthogonal predictors (i.e. number, average item size and total field area) or only the non-numerical dimensions, some of which were correlated with number (i.e. average item size, total field area, total surface area and density). Overall the results obtained with only the orthogonal predictors (i.e. number, average item size and total field area; see *Figure 5—figure supplement 1*) are rather similar to the ones obtained with the full model. Beta weights for number were positive, all along the visual hierarchy, and were attenuated during the size task, yet remaining significantly above zero in all regions (see p-values in *Supplementary file 5*). Beta weights for average item size were almost never significant, while beta estimates for total field area were pronounced predominantly in the earlier visual areas. The most important difference with respect to the full model appears to be a somewhat higher contribution of the number predictor, especially pronounced in early visual regions. Results obtained from the model including only non-numerical dimensions were also similar to the full model (see *Figure 5—figure supplement 2* and p-values in *Supplementary file 6*). Beta weights for average item size were around zero, those for total field area and total surface area were highest in early visual areas and then decreased with the visual hierarchy. Beta weights for density were positive both in early and in higher-level visual areas. The contributions of both total surface area and density show a tendency to be higher here compared to the full model, especially during the number task. However, in both of these analyses it remains ambiguous whether the effects observed for one given predictor are truly driven by that predictor, or potentially attributable to the unmodeled contribution of another one (in the first case, the relatively enhanced effect of number in early visual regions might actually be due to density and TSA which are not included in the model, whereas in the second case, effects attributed to density and TSA could actually be due to the unmodeled contribution of number. Only the most complete model including both numerical and non-numerical dimensions allows us to dissolve this ambiguity.

## Discussion

Our work exploited the enhanced spatial resolution provided by ultra-high field fMRI to reveal how the human brain represents multiple quantitative dimensions of non-symbolic numerical stimuli. Furthermore, we tested whether and at what cortical level the numerical information can be represented and specifically modulated by attention independently of non-numerical visual properties of the image.

At the level of overall regional activity, attending to the numerosity or to the average size of the dots in the array recruited largely overlapping occipital and parietal areas, as also previously observed for perception and comparison of different types of quantities (*Pinel et al., 2004*; *Dormal and Pesenti, 2009*; *Borghesani et al., 2019*); for a meta-analysis on other non-numerical representations see: *Sokolowski et al., 2017*). Two previous fMRI studies have investigated task-related differences in univariate regional activity elicited by numbers of items either in the subitizing (*Leibovich et al., 2015*) or in the estimation (*Leibovich et al., 2016b*) range when participants attended to either the numerosity or to the total surface area of dot arrays while these dimensions varied congruently or incongruently with each other. During comparison of dot arrays within the subitizing range, the overall regional brain activity in the cingulate and right superior frontal gyrus was shown to be modulated by task order (*Leibovich et al., 2015*). When testing larger numerosities, the activation in the right temporal parietal junction was modulated by the congruency between dimensions only when number, but not when total surface area, was the relevant dimension in a comparison task and several regions, including cingulate gyrus, anterior insula, superior frontal gyrus, orbitofrontal and inferior parietal cortex did either respond preferentially during the numerical or the non-numerical task (*Leibovich et al., 2016b*).

While the fact that in the current experiment we do not observe significant differences between tasks at the univariate level might seem to contrast with the results by *Leibovich et al. (2015)* and *Leibovich et al. (2016b)*, several methodological differences are worth noting: first, the non-numerical dimensions investigated are different, being average item size in our case and total surface area in theirs. Second, in the current experiment we made the changes in number and in average item size equally discriminable and we successfully matched task difficulty as demonstrated by the equal

percentage of correct responses across tasks. The numerical and non-numerical dimensions were not matched for perceptual discriminability in the studies by *Leibovich et al. (2015)* and *Leibovich et al. (2016b)*, leaving open the possibility that some of the differences in cortical activation may reflect the higher difficulty associated with the numerical as opposed to the non-numerical task. This possibility seems likely also because the experiments by *Leibovich et al. (2015)* and *Leibovich et al. (2016b)* required participants to perform a comparison on every trial, thus the measured response not only reflects the perceptual process, but also the decision/comparison component. In the current experiment instead, our contrast is related to the sample trials, where participants were only required to perceive/memorize the value on the relevant stimulus dimension, without performing a comparison (which was required only in the occasional match trials). Overall, having balanced task difficulty and excluded the contribution of the comparison process from the response, the current study found no differences between tasks at the univariate level. Only multivariate pattern analysis could detect differences in the way information along the different dimensions was encoded as a function of task in our study.

Multivariate decoding analyses showed that the sample numerosity presented could be read out from brain activity significantly above chance all along the visual stream, however with important differences across regions. When explicitly attended, the numerical information could be read out with gradually higher accuracy following an occipital-parietal gradient, up to a maximum level in the parietal cortices. The effect of attention strongly affected the accuracy of the numerical discrimination in intermediate and higher regions while leaving the accuracy in the early visual areas unaffected. The successful read-out of information related to numerosity from parietal cortices in the current experiment contrasts with some previous studies where fMRI signals discriminative of numerosity could not be detected in the parietal regions (*DeWind et al., 2019*; *Fornaciai and Park, 2018*). Differences in paradigms and sensitivity of the scanners used may account for this discrepancy. Most crucially, in those studies, participants were shown different numerosities and the task required detecting changes in the colour of the dots. Thus, participants' attention may have not been directed to the numerosity of the visual arrays in that case, and the numerical information may have been reduced when focussing on the dots' colour, similarly to what was observed for the size task in the current experiment. Although in the present study we could still read out numerical information even when it was irrelevant for the task, this signal may have remained undetected by less sensitive MRI scanners.

Above chance decoding of numerosity during both tasks was observed here within both the medial IPS parts organized according to visual topography, as well as more lateral/inferior IPS parts outside the visual field maps, to a similar extent. Both regions also allowed to successfully decode the task carried out by the subject, with slightly higher accuracy in the most lateral/inferior IPS part with respect to the most medial IPS part. It has been recently proposed that there might be a subregional specialization along the intra-parietal sulcus, with the most medial/superior regions supporting processing of physical quantities and the lateral/inferior part supporting numerical operations, such as numerical comparison and calculation (*Harvey et al., 2017*; see also *Castaldi et al., 2019*). Our multivariate decoding results only marginally allow us to support distinct roles for these regions, however, addressing this precise question was not an explicit aim of the present study. It remains possible that the representation of numerical information in these different subparts supports different cognitive processes which cannot be differentiated here. For example, while the pattern of activity read out from the IPS field maps might underly the perception of numerosity, the numerical information reflected by the pattern of activity of the more lateral/task-responsive regions of IPS might provide input to internal manipulations of quantity during numerical operations, but further studies are necessary to explore this possibility.

In our study the number presented could be decoded not only in parietal cortex, but already from the earliest stages of visual processing. However, since the multivariate classification analysis collapsed across the non-numerical dimensions of our stimulus set it is unclear whether the information underlying successful decoding was strictly numerical, especially in earlier regions. Some previous studies have dealt with the problem of correlations between numerical and non-numerical stimulus dimensions by controlling for non-numerical features one at the time and testing for fMRI adaptation effects, or replicability of decoding performance or layouts across conditions where individual non-numerical features where controlled for (*Piazza et al., 2004*; *Eger et al., 2009*; *Harvey et al., 2013*; *Harvey and Dumoulin, 2017a*; *DeWind et al., 2019*). When the effects of non-

numerical dimensions were measured directly, this was done in some studies by computing the explained variance or classification performance for each feature in isolation and comparing it to the one for number (*Harvey and Dumoulin, 2017b*; *Cavdaroglu and Knops, 2019*), leaving unspecified the degree to which the simultaneous contribution of several non-numerical dimensions could account for the findings (*Gebuis et al., 2014*). Some other previous studies have taken a different approach, by modeling jointly the effects of numerosity and two non-numerical dimensions (termed 'size in area' and 'spacing') which were designed to be orthogonal to numerosity but do not necessarily constitute natural, perceptually relevant feature dimensions, but rather mathematically defined constructs (*DeWind et al., 2015*; *Park et al., 2015*; *DeWind et al., 2019*; *Fornaciai and Park, 2018*). This design also allowed the authors to estimate, from the combined beta weights of numerosity and the mentioned two orthogonal dimensions, which feature represented by different directions in their stimulus space most accounted for the effects in a given ERP component or brain area. However, brain signals can reflect a combination of responses to multiple quantitative dimensions, and this approach does not permit to distinguish, for example, a modulation by numerosity from two independent modulations by field area and density. In our study, on the contrary, we separated the contributions of numerical and non-numerical stimulus dimensions by applying multiple regression to representational distance matrices which allowed us to test for the extent to which numerosity could explain the pattern of activity while taking into account simultaneously the variability explained by several important natural non-numerical features. Indeed, estimating significantly above zero beta values for number implies that information about numerosity is present in the pattern of activity over and above the contributions of all the other included non-numerical features. We found that information specific to number was detectable beyond the information related to the other dimensions, and that the numerical information was gradually enhanced when progressing along the visual stream when explicitly task relevant, and less strongly represented, although still detectable, when not task-relevant. Importantly, the level of information on other quantitative but non-numerical properties of the image, such as total field area, total surface area and density, although reliably detected especially in earlier brain regions, was not altered when explicitly attending to the numerical quantity. Given that numerical information becomes available in parallel with the other features and independently selectable by attention from early processing stages on, it appears that the human visual system has the capacity to detect signals related to numerosity and separate these from other quantitative dimensions, starting from very basic visual primitives. This supports the existence of a sensory processing mechanisms from which numerosity could be derived directly, rather than making numerical judgements merely indirectly on the basis of percepts of associated non-numerical quantities. Of course, the question of why numerical judgements are nevertheless often influenced by non-numerical quantities, and how these interactions might arise from the involved neuronal populations, remains an important one that deserves further study. Here we modeled the influence of several important natural non-numerical quantitative dimensions on brain activity, though our selection is necessarily non-exhaustive. We acknowledge the fact that we cannot formally rule out that a feature of a different type than those considered by us may have contributed to the effects observed here, and our conclusions hold to the extent of the non-numerical features tested.

The enhancement of numerical information in activation patterns found here when number was the relevant stimulus dimension is extending a growing body of work on the neuronal correlates of feature-based attention. Neurophysiological studies have shown that attention to basic visual features either increases the gain or sharpens responses of neuronal populations preferentially responsive to these features in different visual areas (e.g. *Treue and Martínez Trujillo, 1999*; *McAdams and Maunsell, 2000*; *Reynolds et al., 2000*; *Martinez-Trujillo and Treue, 2004*; *David et al., 2008*), see also: *Carrasco (2011)* for a review). Correspondingly, fMRI decoding studies have found that directing attention to one feature dimension such as orientation, motion direction or color or to particular values within one given dimension improves the read-out of these features from brain activity in early sensory regions (*Kamitani and Tong, 2005*; *Kamitani and Tong, 2006*; *Serences and Boynton, 2007*; *Jehee et al., 2011*) but in some cases also in higher-level areas (*Liu et al., 2011*; *Ester et al., 2016*). According to one influential account, higher-level fronto-parietal areas such as the lateral intraparietal area (LIP) implement spatial 'priority maps' in which the level of activity at individual locations depends jointly on the different features of objects at these locations as well as on top-down factors such as their task relevance, associated reward, etc (*Itti and*

*Koch, 2001*; *Thompson and Bichot, 2005*; *Gottlieb, 2007*; *Sapountzis et al., 2018*). Independent of spatial priority, LIP neurons have also been found to represent higher-level factors such as learned category membership and other non-spatial information (*Freedman and Assad, 2009*) and to flexibly switch between encoding of different visual features, such as color or motion, depending on the task (*Toth and Assad, 2002*; *Ibos and Freedman, 2014*). The idea of a role for intraparietal areas as mere 'priority maps' or reflecting entirely flexible encoding of information on task-relevant features (without intrinsic selectivity) can insufficiently account for our results, since it would predict an equivalent amplification of the representation of average size when this is the attended feature instead of number. This is not what we observed. Our results are thus more compatible with an enhancement of the responses of neuronal populations with intrinsic selectivity to the feature numerosity in these areas (comparable to the one observed for other features in lower-level visual regions).

While the existence of individual neurons tuned to different numbers of items in intraparietal cortex is well established (*Nieder and Miller, 2004*; *Roitman et al., 2007*), the only electrophysiological study that recorded from neurons in the ventral intraparietal (VIP) cortex in macaque monkeys under changing task conditions (*Viswanathan and Nieder, 2015*) found that neurons encoded numerosity to the same extent, regardless of whether the task required to attend to the number or the color of the items. This differs from our results which show a clear attentional amplification of numerosity information. Given that the human IPS 1–5 investigated in the current work is usually considered to be the equivalent of the macaque LIP/VIP complex (*Kastner et al., 2017*), the difference between results may be due to a difference across species, but differences in paradigms and in the nature of the signal recorded in the two studies make it difficult to directly relate the two findings. For example, monkeys were trained initially with the color match to sample task, then re-trained to respond to number, thus implying comparisons across an extended time period and different context, whereas our participants switched between the two tasks within the same scanning session. In addition, it is possible that the color task with a single color per stimulus and a small number of highly distinguishable alternatives placed lower demands on attentional load compared to our average size task, therefore leaving number processing unaltered. Nevertheless, as a common denominator both studies agree on pointing to some degree of spontaneous encoding of numerosity in intraparietal areas under conditions of attention to an orthogonal stimulus dimension.

The gradual enhancement of numerosity information observed by us in the number task when progressing along the dorsal visual stream is compatible with a multi-stage process of the extraction of numerosity where attention may operate at multiple levels over which attentional enhancements accumulate. If numerosity information can be retrieved from multiple levels of the cortical hierarchy, this does not need to imply that this feature is encoded by individual neurons at all these levels, but it may be detectable by multivariate methods even if it existed only in distributed form across the population of neurons. As one speculative interpretation, the numerical information read out from early visual areas could reflect a location map (*Dehaene and Changeux, 1993*), or the process of object segmentation where different individual items start to be separately represented, but this representation may not yet be in a form that is most easily read out for numerical discrimination. Higher areas may progressively transform and concentrate the initially distributed information onto individual neurons, which most likely constitute the base on which we operate when comparing numbers. This interpretation is in line with a recent study showing that although different numerosities could be discriminated based on the pattern of activity in early visual areas and parietal cortex, the behavioral precision of numerical discrimination was correlated with the decoding accuracy only in the latter region (*Lasne et al., 2019*).

While earlier behavioral research suggested that the precision of numerical representation is predictive of formal arithmetic and gets refined with development and mathematical learning (*Halberda and Feigenson, 2008*; *Nys et al., 2013*; *Piazza, 2010*; *Piazza et al., 2010*; *Piazza et al., 2013*), other recent evidence has led to a slightly different view: what might be changing with development and mathematical competence could be the ability to focus on numerical information while filtering out non-numerical dimensions during a numerical comparison task (*Castaldi et al., 2018*; *Piazza et al., 2018*; *Starr et al., 2017*; *Wilkey et al., 2018*). The influence of non-numerical dimensions on numerical judgments decreases over normotypical development and both dyscalculic children (*Bugden and Ansari, 2016*; *Piazza et al., 2018*; *Szucs et al., 2013*; *Wilkey et al., 2018*) and adults (*Castaldi et al., 2018*) seem to be disproportionately affected by non-numerical dimensions during numerical comparison tasks. In light of these behavioral findings, it would be interesting to

see whether in dyscalculic subjects, numerical information at the neuronal level is less precisely encoded overall or merely less accessible to attentional selection. It is possible that the capacity to selectively focus on the numerical information and to enhance it already from early levels of visual analysis on, as shown in the current study, is learned or emerging over development, and future studies should directly test this hypothesis.

A surprising result of the current experiment is that we could not find information about average item size in the pattern of activity in any of the regions examined, even though this feature's perceptual discriminability was equated with the one of numerosity. This suggests that the neural mechanisms supporting average size representation may differ from those engaged during single object size analysis which has been shown to overlap partly with numerosity maps in parietal regions (*Harvey et al., 2015*). Mechanisms for average size perception, and in general for ensemble statistics are still unclear. It has been previously suggested that average item size perception, like density perception, may rely on texture processing mechanisms rather than individual item identification (*Im and Halberda, 2013*). Various regions along the ventral visual stream have been implicated in texture perception. In particular, adaptation studies have identified recovery of fMRI signal in the medial part of the posterior collateral sulcus that was selective for texture as opposed to color or shape of 3D irregular objects (*Cavina-Pratesi et al., 2010*) and the parahippocampal place area (PPA) showed equal release from adaptation for object ensemble and surface textures, suggesting that ensembles and textures are processed similarly (*Cant and Xu, 2012*). It is possible that average size is also represented in the ventral stream which was not covered here, and future studies should focus on these regions to try to detect a representation of average size. What we observed, however, was that beta weights for density obtained from RSA regression became significant in the parietal regions during the size task, suggesting that texture processing mechanisms may be automatically activated during the average size task. This interpretation, however, has to remain speculative and future studies should investigate neural mechanisms relating texture, density and average size processing.

In conclusion, with this study using high-resolution, high-field fMRI we provide direct neuroscientific evidence for a sensory processing mechanism capable of disentangling signals related to visual numerosity from the ones related to associated non-numerical quantities from early stages of cortical processing on, which can then be independently and progressively amplified across the dorsal visual stream when numerical information is explicitly task-relevant. An important goal for the future will be to better understand what are the processing steps and transformations occurring at the different levels of the cortical hierarchy that allow the human brain to isolate numerical information, for example by comparing fMRI data against computational models simulating the visual extraction of numerosity. In addition, it will be important to understand how neuronal representations of numerosity are shaped developmentally and at which cortical levels they can be perturbed to given rise to impaired behavior.

## Materials and methods

### Subjects and MRI acquisition

Twenty healthy adults with normal or corrected vision (10 males and 10 females, mean age 24 years) participated in the study. The study was approved by the regional ethics committee (CPP Ile de France VII, Hôpital de Bicêtre, No. 15–007) and all participants gave written informed consent. Sample size, although not specifically estimated prior to the study, was equal or larger than the one typically used in experiments in the field (see for examples: *Eger et al., 2009*; *Eger et al., 2015*; *Cavdaroglu et al., 2015*; *Castaldi et al., 2016*; *Borghesani et al., 2019*; *Cavdaroglu and Knops, 2019*; *DeWind et al., 2019*; *Fornaciai and Park, 2018*). Functional images were acquired on a SIEMENS MAGNETOM 7T scanner with head gradient insert (Gmax 80mT/m and slew rate 333 T/m/s) and adapted 32-channel head coil (Nova Medical, Wilmington, MA, USA) as T2*-weighted fat-saturation echo-planar image (EPI) volumes with 1.3 mm isotropic voxels using a multi-band sequence (*Moeller et al., 2010*) (https://www.cmrr.umn.edu/multiband/, multi-band [MB] = 2, GRAPPA acceleration with [IPAT] = 2, partial Fourier [PF] = 7/8, matrix = 120×150, repetition time [TR] = 2 s, echo time [TE] = 22 ms, echo spacing [ES] = 0.71 ms, flip angle [FA] = 68°, bandwidth [BW] = 1588 Hz/px, phase-encode direction left >>right). Calibration preparation was done using Gradient Recalled

Echo (GRE) data. Sixty oblique slices covering the occipital, parietal and partially the frontal cortex were obtained in ascending interleaved order. Before the experimental runs two single volumes were acquired with the parameters listed above but with opposite phase encode direction to be used for distortion correction in the later analysis (see Image Processing and Data Analysis). T1-weighted anatomical images were acquired at 0.8 mm isotropic resolution using an MP2RAGE sequence (GRAPPA acceleration with [IPAT] = 3, partial Fourier [PF] = 6/8, matrix = 281×300, repetition time [TR] = 6 s, echo time [TE] = 2.92 ms, time of inversion [TI] 1/2 = 800/2700 ms, flip angle [FA] 1/2 = 4°/5°, bandwidth [BW] = 240 Hz/px,). During scanning participants wore a radiofrequency absorbent jacket (Accusorb MRI, MWT Materials Inc, Passaic, NJ, USA) to minimize so-called 'third-arm' or 'shoulder' artifacts due to regions where the head gradient is unable to unambiguously spatially encode the image (*Wald et al., 2005*). Head movement was minimized by padding and tape. Visual stimuli were back-projected onto a translucent screen at the end of the scanner bore and viewed through a mirror attached to the head coil. Participants held two response buttons in their left and right hands.

## Stimuli and procedure

During fMRI scanning participants were centrally presented with heterogeneous arrays of dots, half black, and half white, on a mid-gray background to ensure that total luminance was not a cue for number, a strategy used in many previous studies (*Anobile et al., 2012*; *Anobile et al., 2014*; *Anobile et al., 2016c*; *Anobile et al., 2016a*; *Anobile et al., 2018*; *Cicchini et al., 2016*; *Dakin et al., 2011*; *Fornaciai et al., 2016*; *Morgan et al., 2014*; *Ross, 2010*).

The generated sets of dots were orthogonally varied in number, average item size and total field area for a total of 18 conditions: six, ten or seventeen dots were presented with either small, medium or large average item area (0.04, 0.07, 0.12 visual squares degree) and designed to fall within a small or large total field area (defined by a virtual circle of either about 5 or 7.5 visual degree diameter). This implies that higher numbers were associated with higher total surface areas (total surface area for number six, ten and seventeen respectively corresponded to: 0.25, 0.42, 0.71 $vd^2$ for the smallest average item size, to 0.42, 0.71 and 1.19 $vd^2$ for the medium average item size and to 0.72, 1.19 and 2.03 vd2 for the largest average item size, correlation between numerosity and total surface area: rho = 0.68, p = 0.002, see *Figure 1—figure supplement 1A*) and higher density (density for numerosity six, ten and seventeen respectively corresponded to: 0.30, 0.50 and 0.85 dots/$vd^2$ for the small total field area and to: 0.14, 0.23, 0.39 dots/$vd^2$ for the large total field area, correlation between numerosity and density: rho = 0.71, p = 0.0008, see *Figure 1—figure supplement 1B*). Despite the significant correlations between numerical and non-numerical dimensions, some pairs of stimuli had equal or similar values of total surface area and density across numerosities and sizes (for example the array of ten dots with the smallest average item size had total surface area equal to 0.42 $vd^2$ which corresponded to the total surface area of the array of six dots with medium average item size, see *Figure 1—figure supplement 1* for a more comprehensive visualization of the full stimuli set). Convex hull was not explicitly controlled and a-posteriori calculation showed that it was correlated with number (average convex hull for numerosity six, ten and seventeen respectively corresponded to: 9, 14 and 18 $vd^2$ for the small total field area and to: 20, 31, 41 $vd^2$ for the large total field area, correlation between numerosity and convex hull: rho = 0.57, p = 0.01) and total field area (correlation between total field area and convex hull: rho = 0.78, p = 0.0001). Numbers and average item sizes were chosen to be perceptually equally discriminable based on a previous behavioral study (*Castaldi et al., 2018*). Total field areas were chosen so that arrays of dots could be sufficiently sparse (~1 dot/$vd^2$) to target the 'number regime' (*Anobile et al., 2014*; *Anobile et al., 2015*).

Within each run participants performed two tasks in different blocks, as indicated by the written task instructions provided at the beginning of each block. Instructions were shown for 2 s and specified whether participants had to attend either to the number of dots (number task) or to the average item size of the dots (size task) in the array. Six seconds after the instruction a delayed comparison task started with brief presentation (500 ms) of a sample dot array stimulus. At each trial participants attended to the cued dimension of the sample stimulus and held this information in memory until the following trial was presented, knowing that a comparison response with the following trial may be required. After a variable ISI of 3.5–5.5 s, a second dot array was presented. If the color of the fixation point remained unchanged (green), no comparison was required and participants only had to

update their memory with the new sample stimulus. If instead the fixation point changed color (turning to red 1 s before the stimulus presentation) participants had to compare the current stimulus (match stimulus) with the one held in memory and decide whether the current stimulus was larger or smaller (on the attended dimension) than the previous one. Response was provided by button press and after 5.5 s the next sample stimulus was presented and the whole procedure started again. Match stimuli were designed to be ~2 JNDs larger or smaller than the previously presented sample stimulus on the attended dimension, based on each participant's Weber fraction as measured in a behavioral test prior to the fMRI scanning, while the unattended dimension was the same as the previous sample stimulus.

Twenty trials were presented in each block: one trial for each one of the 18 sample stimulus conditions (3 numerosity x 3 sizes x two total field areas) and two match trials. The hands assigned to either the 'smaller' or 'larger' response were inverted in the middle of the scanning session, that is after the third run, and counterbalanced across subjects. Within the scanning session participants performed six runs of ~7 min and 44 s. Each run included four blocks where the two tasks alternated. The type of task with which the run started was balanced across runs and participants.

To measure their numerical and average size acuity, participants performed a behavioral test prior to the fMRI scanning. In different sessions participants were shown two consecutive centrally presented arrays of dots and were required to perform a discrimination task on the attended dimension (either numerosity or average item size) by pressing the left or the right arrow (to choose the first or the second stimulus respectively). The set of stimuli used included arrays of 5,7,9,11,15 and 20 dots (ratios 0.5, 0.7, 0.9, 1.1, 1.5 and 2 with respect to the reference of 10 dots) that could be displayed with the average dot areas of 0.05, 0.06, 0.08, 0.11, 0.15 and 0.2 visual square degrees (ratios 0.5, 0.6, 0.8, 1.1, 1.5 and 2 with respect to the reference of 0.1 visual square degrees). Dots were randomly drawn within two possible virtual circles of ~5.8 and 7.6 visual degrees diameter. Reference and test stimuli could appear either as first or as second stimulus. After task instructions and twelve practice trials, participants performed three sessions of one task and three sessions of the other, with counterbalanced order across subjects. For each task participants performed a total of 432 comparisons (6 numerosities x six average item sizes x two total field areas x two presentation order x three sessions). To quantify participants' precision in number and size judgments, we computed the JND for each task. The percentage of test trials with 'greater than reference' responses was plotted against the log-transformed difference between test and reference and fitted with a cumulative Gaussian function using Psignifit toolbox (*Schütt et al., 2016*). The difference between the 50% and the 75% points yielded the JND.

Stimuli and paradigms were generated and presented under Matlab 9.0 using PsychToolbox routines (*Brainard, 1997*).

## Image processing and data analysis

EPI images were motion-corrected and co-registered to the first single band reference image using statistical parametric mapping software (SPM12, https://www.fil.ion.ucl.ac.uk/spm/software/spm12/). The single-band reference images of the two initial volumes acquired with opposite phase encode directions served to estimate a set of field coefficients using topup in FSL (https://fsl.fmrib.ox.ac.uk/fsl/fslwiki/FSL), which was subsequently used to apply distortion correction (apply_topup) to all EPI images. Cortical surface reconstruction and boundary based registration of single band reference images to each subject's cortical surface, as well as a minimal amount of surface constrained smoothing (FWHM = 1.5 mm) for noise reduction were performed in Freesurfer (https://surfer.nmr.mgh.harvard.edu/).

The preprocessed EPI images (in subjects' native space) were entered into a general linear model separately modeling the effects of the 36 sample conditions (3 numerosities x three average item sizes x two total field areas x two tasks, within each run the two repetitions for each condition were pooled together), the match stimulus separately for left and right hand and the written instructions at the beginning of the block as stick functions (using the default of 0 duration for events) convolved with the standard hemodynamic response function. The six motion parameters were included in the GLM as covariate of no interest. An AR(1) model was used to account for serial auto-correlation and low-frequency signal drifts were removed by a high-pass filter with a cutoff of 192 s. In each subject we contrasted the activation elicited by: all the sample stimuli during the number tasks against the implicit baseline (contrast name: 'Judge Number > Baseline'); all the sample stimuli during the size

tasks against the implicit baseline (contrast name: 'Judge Size >Baseline'); all the sample stimuli during the number tasks against all the sample stimuli during the size tasks (contrast name: 'Judge Number > Judge Size). After creating the contrasts in each single subject's volume space, the contrast images were projected onto the surface with Freesurfer, aligned to fsaverage and smoothed with a 3 mm fwhm Gaussian kernel. The second-level group analysis was then performed in the surface space.

The beta estimates for the sample stimulus conditions from the first-level analysis (one beta estimate per run and condition) were entered into pattern recognition analysis. In each subject we defined anatomical regions of interest (ROIs) derived from a surface based probabilistic atlas (*Wang et al., 2015*) where regions are defined based on retinotopy. ROIs for V1 to IPS5 were created on the Freesurfer surface and projected back into each subject's volume space. For each ROI we merged the left and right hemisphere. ROIs were further merged into three large ROIs corresponding to early (V1 to V3), intermediate (V3A, V3B and V7, also known as IPS0) and higher-level (IPS one to IPS5) areas. In addition we focused the analysis on individual regions: V1, V2, V3, V3AB (merging V3A and V3B), V7, IPS12 (merging IPS 1 and 2), IPS345 (merging IPS 3, 4 and 5). Finally, for supplementary analyses, we defined a region along the intraparietal sulcus excluding the field map representation IPS 0–5. This region was defined by excluding V7 (also called IPS0) and IPS1-5 from the intraparietal and transverse parietal sulci ROI as defined by the *Destrieux et al. (2010)*. Within each one of these bilateral regions we selected on a subject-by-subject basis an equal number of 800 voxels that responded most strongly to the orthogonal contrast 'all sample stimuli > baseline' for pattern recognition analysis. To evaluate the degree of spatial consistency of the selected voxels across subjects we created an overlap map with Freesurfer (*Figure 3B*): single subjects' ROIs were aligned to fsaverage and the number of subjects for which a given location was included in their specific ROI was represented by a heat map (with yellow color meaning that a given location was selected in all subjects).

Pattern classification analysis was performed in scikit-learn (*Pedregosa et al., 2011*) using beta estimates after subtracting the voxel-wise mean across conditions by applying linear support vector machines (SVM) with regularization parameter C = 1. Classification analysis was performed leaving patterns of one run out at each loop of the 6-fold cross-validation cycle. This implies that classifiers were trained on five betas per condition and tested with the left-out beta images (one per condition). The classification accuracies obtained for each cycle were then averaged together. Pairwise classification was performed for all pairs of numerosities collapsing across the size and total field area dimensions, but keeping patterns separated by task. Classification accuracy was then averaged across all pairs of numerosities for each task. A one-sample t-test against the theoretical chance level of 50% was performed to evaluate significance of discrimination. Repeated measures ANOVAs where then performed on classification accuracies with ROI and task as factors. For the results described in the Supplementary Material, equivalent analyses were performed on the decoding accuracies when the classifier was trained and tested to discriminate between tasks.

For representational similarity analysis (*Kriegeskorte, 2008*; *Kriegeskorte and Kievit, 2013*) the GLM was performed concatenating the runs and obtaining one single beta per condition, task and subject. Comparable to the procedure of the pattern classification analysis, voxel-wise scaling was applied by subtracting the mean across conditions. Neural representational dissimilarity matrices (neural RDMs) for each task and ROI were created by computing the correlation distance (1 – the Pearson correlation across voxels) between activity patterns associated with all possible pairs of conditions using CoSMoMVPA Toolbox (*Oosterhof et al., 2016*). The neural RDMs were then entered in a multiple regression with five predictors corresponding to matrices encoding the distance between all pairs of conditions on a logarithmic scale for the different quantitative dimensions defining the dot arrays: number, average item size, total field area, total surface area and density. To explore potential effects of correlations between predictors, equivalent supplementary analyses included only orthogonal dimensions (i.e. number, average item size and total field area), or all non-numerical dimensions except number (i.e. average item size, total field area, total surface area and density). In the multiple regression analysis all distance matrices were z-transformed before estimating the regression coefficients. The obtained beta weights for each dimension and ROI were tested with one-sample t-tests against zero across subjects. The effects of ROI, dimension and task were analyzed with repeated measures ANOVAs.

## Acknowledgements

This work was funded by the French National Research Agency (grant No ANR-14-CE13- 0020–01 to E Eger). We thank F De Martino and V Kemper for advice on fMRI acquisition parameters and procedures.

## Additional information

### Funding

| Funder | Grant reference number | Author |
|---|---|---|
| Agence Nationale de la Recherche | ANR-14-CE13-0020-01 | Evelyn Eger |

The funders had no role in study design, data collection and interpretation, or the decision to submit the work for publication.

### Author contributions

Elisa Castaldi, Conceptualization, Data curation, Software, Formal analysis, Validation, Investigation, Visualization, Methodology, Writing—original draft, Writing—review and editing; Manuela Piazza, Conceptualization, Supervision, Validation, Visualization, Methodology, Writing—review and editing; Stanislas Dehaene, Conceptualization, Supervision, Validation, Methodology, Writing—review and editing; Alexandre Vignaud, Validation, Investigation, Methodology, Writing—review and editing; Evelyn Eger, Conceptualization, Resources, Data curation, Software, Formal analysis, Supervision, Funding acquisition, Validation, Investigation, Visualization, Methodology, Writing—original draft, Project administration, Writing—review and editing

### Author ORCIDs

Elisa Castaldi (iD) https://orcid.org/0000-0003-0327-6697

### Ethics

Human subjects: The study was approved by the regional ethics committee (CPP Ile de France VII, Hôpital de Bicêtre, No. 15-007) and all participants gave written informed consent.

### Decision letter and Author response

Decision letter https://doi.org/10.7554/eLife.45160.023
Author response https://doi.org/10.7554/eLife.45160.024

## Additional files

### Supplementary files

• Supplementary file 1. Statistical results for the performance of the classifiers trained to discriminate between different numerosities. The table reports t-values, degrees of freedom (Dof), p-values and confidence intervals of the two-tailed t-tests against 0.5 (chance level) used to evaluate the accuracies of number classification for every ROI and task.
DOI: https://doi.org/10.7554/eLife.45160.013

• Supplementary file 2. Statistical results for beta weights obtained from the RSA multiple regression. The table shows t-values, degrees of freedom (Dof), p-values and confidence intervals of two-tailed t-tests against zero across subjects for every ROI and dimension (N: number, S: average item size, TFA: total field area, TSA: total surface area, D: density) for the number (left table) and size (right table) tasks.
DOI: https://doi.org/10.7554/eLife.45160.014

• Supplementary file 3. Statistical results for the performance of the classifiers trained to discriminate between tasks. The table reports t-values, degrees of freedom (Dof), p-values and confidence

intervals of the two-tailed t-tests against 0.5 (chance level) used to evaluate the significance of task classification for every ROI.

DOI: https://doi.org/10.7554/eLife.45160.015

• Supplementary file 4. Statistical results for the ROI IPS excluding IPS 0–5. (**a**) Statistical results for the performance of the classifiers trained to discriminate between numerosities during the number (left table) and size (right table) task for the ROI IPS excluding IPS 0–5. The table reports the statistical results of the two-tailed t-tests against 0.5 (chance level). (**b**). Statistical results for the performance of the classifiers trained to discriminate between tasks for the ROI IPS excluding IPS 0–5. The table reports the statistical results of the two-tailed t-tests against 0.5 (chance level). (**c**) Statistical results for beta weights obtained from the RSA multiple regression for the ROI IPS excluding IPS 0–5. The table shows t-values, degrees of freedom (Dof), p-values and confidence intervals of two-tailed t-tests against zero across subjects for every dimension (N: number, S: average item size, TFA: total field area, TSA: total surface area, D: density) for the number (left table) and size (right table) tasks.

DOI: https://doi.org/10.7554/eLife.45160.016

• Supplementary file 5. Statistical results for beta weights obtained from the RSA multiple regression when including only number, average item size and total field area as regressors. The table shows t-values, degrees of freedom (Dof), p-values and confidence intervals of two-tailed t-tests against zero across subjects for every ROI and dimension (N: number, S: average item size, TFA: total field area) for the number (left table) and size (right table) tasks.

DOI: https://doi.org/10.7554/eLife.45160.017

• Supplementary file 6. Statistical results for beta weights obtained from the RSA multiple regression when including only non-numerical dimensions as regressors (i.e., average item size, total field area, total surface area and density). The table shows t-values, degrees of freedom (Dof), p-values and confidence intervals of two-tailed t-tests against zero across subjects for every ROI and dimension (S: average item size, TFA: total field area, TSA: total surface area, D: density) for the number (left table) and size (right table) tasks.

DOI: https://doi.org/10.7554/eLife.45160.018

• Transparent reporting form

DOI: https://doi.org/10.7554/eLife.45160.019

### Data availability

Individual subjects' data points for behavioural and fMRI results for all regions of interest, corresponding to Figures 2A, 3C, 5, Figure 3—figure supplements 1 and 2, and Figure 5—figure supplements 1 and 2 as. cvs files. The maps displayed in figure 2B-D and 3B are provided in a format readable with Freesurfer/Freeview, one of the most widely used free neuroimaging softwares. The functional imaging dataset is available via the Open Science Framework (osf.io/6zch2).

The following dataset was generated:

| Author(s) | Year | Dataset title | Dataset URL | Database and Identifier |
|---|---|---|---|---|
| Castaldi E, Eger E | 2019 | Data from: Attentional amplification of neural codes for number independent of other quantities along the dorsal visual stream | https://osf.io/6zch2 | Open Science Framework, 6zch2 |

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
