## [Decision Letter]

Thank you for submitting your article "Attentional amplification of neural codes for number independent of other quantities along the dorsal visual stream" for consideration by *eLife*. Your article has been reviewed by three peer reviewers, including Daniel Ansari as the Reviewing Editor and Reviewer #1, and the evaluation has been overseen by Joshua Gold as the Senior Editor. The following individual involved in review of your submission has agreed to reveal their identity: Ben Harvey (Reviewer #2).

The reviewers have discussed the reviews with one another and the Reviewing Editor has drafted this decision to help you prepare a revised submission.

Summary:

This manuscript reports the result of an experiment that addresses how numerosity is extracted in the brain and examines whether and at what level such numerosity extraction is independent of other visual variables that co-vary with numerosity (e.g. overall area). Participants' attention was either directed to the numerosity of a dot array or average item size. The results reveal that there is increasing sensitivity to numerosity the further away one moves from primary visual cortex and numerosity sensitivity is particularly strong in the IPS.

The reviewers all agreed that this is a well-written manuscript describing an elegantly designed experiment with sophisticated data acquisition and analysis methods.

Essential revisions:

1).Against the background of the data, is the conclusion that the findings "reveal a dedicated extraction mechanism for numerosity that operates independently of other quantitative dimensions of the stimuli" justified? The reviewers wondered whether such a strong statement is warranted against the background of the evidence presented. More specifically, just because numerosity can be decoded in higher-level regions when it is being attended to and explains unique variance this does not exclude the possibility that whatever representation is being activated there is the product of the integration of the non-numerical variables. It may just be a more abstract representation that emerged over the course of development or that such a representation is actively being constructed given the attentional focus and task demand. You need to explain in more detail how they can reject the alternative account and instead conclude that there is a 'dedicated' mechanism for numerosity processing. The fact that numerosity explains unique variance in the multiple regression analyses does not mean that it is not an emergent property?

2) Related point 1, you state that: "The fact that such specifically numerical information is found from early stages of the cortical hierarchy on, and that attentional modulation does not affect associated non-numerical quantities makes it unlikely that numerical judgements would only be made indirectly on the basis of different non-numerical features. " – yes it makes it unlikely but this explanation cannot be fully excluded on the basis of the data, especially because not all possible non-numerical dimensions that have been shown to influence numerosity processing were excluded and because the classification accuracies of numerosity were not high (and non-significant) in the early stages of the cortical hierarchy.

3) Other fMRI studies (at least two) have compared neural activation of non-symbolic dot stimuli with the instruction to either respond based on size or based on numerosity but are omitted from the background literature. How do you reconcile the current null results when contrasting activity related to the two tasks with previous results showing task-related differences?

Leibovich, Henik and Salti, 2015; Leibovich et al., 2016.

4) The stimuli description emphasizes the orthogonality of stimulus visual parameters and the "partial decorrelation" of numerosity and density. However, based on a rough calculation of stimulus characteristics from Figure 1A, total surface area and numerosity can be estimated to correlate at about r = 0.66, which can be considered a strong correlation. These numbers was calculated by multiplying the number of dots by the average item area and then computing a bivariate correlation between the list of stimulus numerosities and total surface areas.

The strength of this correlation is difficult to tell from the example RDM's presented in Figure 4, but remains quite strong. Rather than de-emphasize this potential confound as a "partial decorrelation" and a "good dissociation", the authors should indicate that total surface area and numerosity remain highly correlated despite orthogonality along other stimulus parameters. This is a very serious caveat to all analyses and results in the current study and should be emphasized accordingly. The same may be true of density. Similar calculations should be run and included. Further, these correlations may be affecting the results of the multiple regression, as highly correlated predictors often do. It may be appropriate to calculate variance inflation factors if the correlations are high enough. Further, this issue of independent contributions to variance in the multiple regression model may be explored through removing and adding back in variables that may be correlated. For example, does total surface area or item size show a different pattern of results when number is removed from the model?

[Editors' note: further revisions were requested prior to acceptance, as described below.]

Thank you for resubmitting your work entitled "Attentional amplification of neural codes for number independent of other quantities along the dorsal visual stream" for further consideration at *eLife*. Your revised article has been favorably evaluated by Joshua Gold as the Senior Editor, a Reviewing Editor, and two reviewers.

As will see from their comments below, both reviewers feel that your manuscript has been substantially improved and feel that it is not ready to be accepted for publication. Before doing so, I would ask you to address the remaining comments by reviewer 3 (see below):

*Reviewer #2:*

The authors have addressed all of my concerns.

*Reviewer #3:*

The authors have addressed my concerns in their responses and additional analyses. I have two suggestions that do not need further review in order for the manuscript to be considered acceptable for publication.

First, it would be helpful if all supplementary files were converted to a standard, non-proprietary format. This is most important for beta-weight matrices or any files that a general reader may want to read. These could easily be saved as a.csv file. That said, it seems reasonable that stimulus design files or complex files like 3-dimensional matrices necessary for a replication could remain as. mat files.

Second, I find the schematic for the "Neural RDM" very helpful (the one with acronyms for each row and column). I would suggest adding it to the figure in the main paper along with a caption that denotes those abbreviations. It will clear up considerable confusion for the reader.

Overall, this is an excellent study. Well done.

---

## [Author Response]

Essential revisions:1) Against the background of the data, is the conclusion that the findings "reveal a dedicated extraction mechanism for numerosity that operates independently of other quantitative dimensions of the stimuli" justified? The reviewers wondered whether such a strong statement is warranted against the background of the evidence presented. More specifically, just because numerosity can be decoded in higher-level regions when it is being attended to and explains unique variance this does not exclude the possibility that whatever representation is being activated there is the product of the integration of the non-numerical variables. It may just be a more abstract representation that emerged over the course of development or that such a representation is actively being constructed given the attentional focus and task demand. You need to explain in more detail how they can reject the alternative account and instead conclude that there is a 'dedicated' mechanism for numerosity processing. The fact that numerosity explains unique variance in the multiple regression analyses does not mean that it is not an emergent property?

We have the impression that one important result of our study was not properly received, and we apologize if it was not clearly presented. We would like to stress the fact that the numerical information is present in our data already at early levels (as indicated by both the significant decoding accuracy and RSA beta values for number starting already from early visual cortex on, see Supplementary file 1 and 2 for significance values in individual areas), and this information remained significant also when attention was directed away from the numerical dimension. In other words, the numerical information is contained in the pattern of activity already in the early visual areas, and not only in higher level regions.

It is not entirely clear to us what the reviewers exactly refer to by a ‘product of integration’ and how and at what level in the brain this could be assumed to happen (e.g., at sensory extraction or decision stages). The fact that information related to number and the other properties becomes available in parallel at early levels seems to suggest to us that numerical information can be derived from very basic visual features during sensory processing, without the need to first construct explicit precepts of density and field area for example, based on which number would only be inferred indirectly.

We certainly agree with the reviewers that the capacity to selectively focus on the numerical information and to enhance it already at early levels as we show here, could be learned and/or may have emerged over development. We have now added a paragraph mentioning that the ability to focus on number may be learned/emerge over development in the Discussion (tenth paragraph). We now also avoid using the formulation ‘a dedicated extraction mechanism for numerosity’ throughout the manuscript, to not give the impression that we have evidence for an innate mechanism subserved by ‘dedicated’ neurons in the sense that they would be exclusively concerned with processing number and nothing else. Instead we replaced such formulation with’ a sensory extraction mechanism yielding information on numerosity separable from other dimensions’ or with similar expressions, such as ‘a neuronal mechanism directly sensitive to the numerosity of visual sets’ or ‘a sensory processing mechanism capable of disentangling signals related to visual numerosity from the ones related to associated non-numerical quantities’.

2) Related point 1, you state that: "The fact that such specifically numerical information is found from early stages of the cortical hierarchy on, and that attentional modulation does not affect associated non-numerical quantities makes it unlikely that numerical judgements would only be made indirectly on the basis of different non-numerical features. " – yes it makes it unlikely but this explanation cannot be fully excluded on the basis of the data, especially because not all possible non-numerical dimensions that have been shown to influence numerosity processing were excluded and because the classification accuracies of numerosity were not high (and non-significant) in the early stages of the cortical hierarchy.

As mentioned in point 1, we would like to point out that the classification accuracies (as well as results for number in the multiple regression RSA) were statistically significant already at the early stages of the cortical hierarchy, as detailed in the text and in Supplementary file 1.

In the revised version of the manuscript, we have replaced the formulation cited above by:

‘Given that numerical information becomes available in parallel with the other features and independently selectable by attention from early processing stages on, it appears that the human visual system has the capacity to detect signals related to numerosity and separate these from other quantitative dimensions, starting from very basic visual primitives. […] We acknowledge the fact that we cannot formally rule out that a feature of a different type than those considered by us may have contributed to the effects observed here, and our conclusions hold to the extent of the nonnumerical features tested.’

3) Other fMRI studies (at least two) have compared neural activation of non-symbolic dot stimuli with the instruction to either respond based on size or based on numerosity but are omitted from the background literature. How do you reconcile the current null results when contrasting activity related to the two tasks with previous results showing task-related differences?Leibovich, Henik, and Salti, 2015; Leibovich et al., 2016.

We thank the reviewers for pointing out these two studies that are now mentioned in a dedicated paragraph in the Discussion (second and third paragraphs). We believe that these two studies differ in several methodological aspects which might have at least in part contributed to the different results obtained. First, differently from Leibovich et al., 2015, 2016b, we asked participants to compare the average item size, rather than the total surface area of the dot arrays. Moreover, we matched task difficulty for comparing item size with the one for comparing numerosity. We have noticed that task difficulty was not successfully matched in the Leibovich et al. studies, which reported differences in accuracy and reaction times across tasks. Differences in task difficulty might have led to unequal cognitive load which might have been detected by the univariate contrast between tasks. Finally, we noticed that Leibovich et al. required participants to perform an explicit comparison one very trial, therefore adding a decisional component to the brain response which is instead not present in our analyses (where the comparison was done only during the occasional match trials which were not included in the contrast reported).

4) The stimuli description emphasizes the orthogonality of stimulus visual parameters and the "partial decorrelation" of numerosity and density. However, based on a rough calculation of stimulus characteristics from Figure 1A, total surface area and numerosity can be estimated to correlate at about r = 0.66, which can be considered a strong correlation. These numbers was calculated by multiplying the number of dots by the average item area and then computing a bivariate correlation between the list of stimulus numerosities and total surface areas.The strength of this correlation is difficult to tell from the example RDM's presented in Figure 4, but remains quite strong. Rather than de-emphasize this potential confound as a "partial decorrelation" and a "good dissociation", the authors should indicate that total surface area and numerosity remain highly correlated despite orthogonality along other stimulus parameters. This is a very serious caveat to all analyses and results in the current study and should be emphasized accordingly. The same may be true of density. Similar calculations should be run and included. Further, these correlations may be affecting the results of the multiple regression, as highly correlated predictors often do. It may be appropriate to calculate variance inflation factors if the correlations are high enough. Further, this issue of independent contributions to variance in the multiple regression model may be explored through removing and adding back in variables that may be correlated. For example, does total surface area or item size show a different pattern of results when number is removed from the model?

We have now added a more detailed description of the stimuli characteristics in the Materials and methods section and in addition report the values of correlations between numerical and nonnumerical dimensions across the 18 conditions.

The aim of the current study was not to create a stimulus set which would control for all the nonnumerical features at the same time (or decorrelate number from all non-numerical dimensions), which is impossible to do. Rather, the approach used here was to include predictors for numerical and multiple non-numerical dimensions into a multiple regression model, thus estimating beta weights for every dimension which reflect the contribution of each given dimension above and beyond what all the other predictors can explain.

Generally, in the presence of correlations between predictors in a multiple regression, the amount of variance uniquely explained by each predictor is reduced, and betas have a tendency to be less reliably estimated (in the extreme case of complete collinearity they would be completely inestimable). Therefore, correlations between predictors increase the likelihood of non-significant results, since due to the smaller effect size and less reliable estimation, multiple independent replications of the same analysis (such as over subjects) are unlikely to yield a systematic effect. On the contrary, every significant effect of one predictor (as we observe here across subjects) obtained in spite of the correlations is valid and can be thought of as reflecting only the variance that is uniquely explained by that given predictor. For a discussion related to this point, see also for example: http://imaging.mrccbu.cam.ac.uk/imaging/DesignEfficiency#Correlation_between_regressors).

To get an impression of how much less reliably beta weights may have been estimated in our model due to the correlations between predictors, as suggested by the reviewers, we computed variance inflation factors for all predictors, and now report them also in the manuscript for information purposes. The resulting values were: 1.4874 for number, 1.1957 for average item size, 1.2048 for total field area, 1.3238 for total surface areas and 1.4591 for density. All of these point to an only moderately increased variance in the estimation, well below the threshold for what is typically considered excessive collinearity (10 or 5, according to different sources). This is likely the reason why we can still obtain the significant group results reported for the different predictors in our study.

As requested by the reviewers, we have also explored the effects of removing vs. including specific predictors in our model (Figure 5—figure supplement 1, Supplementary file 5; Figure 5—figure supplement 2, Supplementary file 6). In particular, in a first model we have carried out multiple regression analyses including only the predictors that orthogonally varied with number (i.e. average items size and total field area, see Figure 5—figure supplement 1, Supplementary file 5) and excluding the correlated ones (i.e. total surface area and density). Furthermore, in a second model we have computed the RSA multiple regression including only the non-numerical predictors (both those correlated with and those orthogonal to number, see Figure 5—figure supplement 2, Supplementary file 6).

We have added a paragraph in the Results section (subsection “Models with reduced number of predictors”) reporting the outcome of these analyses and figures and statistics are reported in the supplementary material (Figure 5—figure supplement 1, Supplementary file 5; Figure 5—figure supplement 2, Supplementary file 6).

We found that in the model including only the orthogonal dimensions (i.e. number, average item size and total field area), the triple interaction between ROI, task and dimension was significant both for the three large regions: F(2.86,54.43)=4.73, p=0.006 as well as for the individual regions: F(4.52,85.95)=3.34, p=0.01. Overall, early visual betas for number appeared somewhat enhanced in this case compared to the original model including all predictors. On the contrary, when excluding number from the model, but including all non-numerical dimensions (including those that in the full design are partially correlated with number), the triple interaction between ROI, task and dimension was significant neither for the three large regions: (F(4.12,78.24)=1.107, p=0.36) nor for the individual regions (F(5.66,107.50)=0.883, p=0.50). The interaction between task and dimension was significant both for the three large regions (F(3.12,40.42)=3.65, p=0.03) as well as for the individual regions (F(2.07,39.39)=3.32, p=0.04). In this case, the effect appeared to be mainly driven by total surface area and density yielding somewhat higher betas than in the original full model, most pronounced during the number task. The concern with both of these analyses is that it remains ambiguous whether the effects observed for one given predictor are truly driven by that predictor, or potentially attributable to the unmodeled contribution of another stimulus dimension (in the first case, the relatively enhanced effect of number in early visual regions might actually be due to density and TSA which are not included in the model, whereas in the second case, some effects attributed to density and TSA could actually be due to the unmodeled contribution of number.

Only our initial, most complete model reported in the manuscript is able to resolve this ambiguity, and we believe it is best suited to address our main question, which was to ask whether there is a unique contribution of numerosity to activation patterns above and beyond what can be explained by the ensemble of the non-numerical dimensions.

[Editors' note: further revisions were requested prior to acceptance, as described below.]

As will see from their comments below, both reviewers feel that your manuscript has been substantially improved and feel that it is not ready to be accepted for publication. Before doing so, I would ask you to address the remaining comments by reviewer 3 (see below):Reviewer #3:The authors have addressed my concerns in their responses and additional analyses. I have two suggestions that do not need further review in order for the manuscript to be considered acceptable for publication.First, it would be helpful if all supplementary files were converted to a standard, non-proprietary format. This is most important for beta-weight matrices or any files that a general reader may want to read. These could easily be saved as a.csv file. That said, it seems reasonable that stimulus design files or complex files like 3-dimensional matrices necessary for a replication could remain as. mat files.

All matrices containing the behavioral performance, decoding accuracies and betaweights, previously provided as. mat files, are now shared in. cvs format

Second, I find the schematic for the "Neural RDM" very helpful (the one with acronyms for each row and column). I would suggest adding it to the figure in the main paper along with a caption that denotes those abbreviations. It will clear up considerable confusion for the reader.

The schematic for the neural RDM with labels has been added to Figure 4 in the main text and the caption has been updated accordingly.